# Enteric glia as a source of neural progenitors in adult zebrafish

**Sarah McCallum[1], Yuuki Obata[1], Evangelia Fourli[1], Stefan Boeing[2], Christopher J Peddie[3], Qiling Xu[4], Stuart Horswell[2], Robert N Kelsh[5], Lucy Collinson[3], David Wilkinson[4], Carmen Pin[6], Vassilis Pachnis[1]\*, Tiffany A Heanue[1]\***

[1]Development and Homeostasis of the Nervous System Laboratory, The Francis Crick Institute, London, United Kingdom; [2]Bionformatics & Biostatistics Science Technology Platform, The Francis Crick Institute, London, United Kingdom; [3]Electron Microscopy Science Technology Platform, The Francis Crick Institute, London, United Kingdom; [4]Neural Development Laboratory, The Francis Crick Institute, London, United Kingdom; [5]Department of Biology and Biochemistry, University of Bath, Bath, United Kingdom; [6]Clinical Pharmacology and Quantitative Pharmacology, Clinical Pharmacology and Safety Sciences, R&D, AstraZeneca, Cambridge, United Kingdom

**Abstract** The presence and identity of neural progenitors in the enteric nervous system (ENS) of vertebrates is a matter of intense debate. Here, we demonstrate that the non-neuronal ENS cell compartment of teleosts shares molecular and morphological characteristics with mammalian enteric glia but cannot be identified by the expression of canonical glial markers. However, unlike their mammalian counterparts, which are generally quiescent and do not undergo neuronal differentiation during homeostasis, we show that a relatively high proportion of zebrafish enteric glia proliferate under physiological conditions giving rise to progeny that differentiate into enteric neurons. We also provide evidence that, similar to brain neural stem cells, the activation and neuronal differentiation of enteric glia are regulated by Notch signalling. Our experiments reveal remarkable similarities between enteric glia and brain neural stem cells in teleosts and open new possibilities for use of mammalian enteric glia as a potential source of neurons to restore the activity of intestinal neural circuits compromised by injury or disease.

**\*For correspondence:**
vassilis.pachnis@crick.ac.uk (VP);
tiffany.heanue@crick.ac.uk (TAH)

**Reviewing editor:** Julia Ganz,

## Introduction

Tissue integrity and repair depend on the regulated dynamics of adult stem cells, which share the capacity to replenish cellular compartments depleted by physiological turnover or disease. Studies on neural stem cells (NSCs) have advanced fundamental brain research and opened new and exciting opportunities for regenerative neuroscience (*Morales and Mira, 2019*). However, as NSC research has focused primarily on the central nervous system (CNS), our understanding of the homeostasis and regenerative potential of peripheral neural networks, and particularly the enteric nervous system (ENS), is minimal and at best phenomenological. This gap in knowledge impedes progress in fundamental gastrointestinal biology and stymies the development of potential therapeutic strategies for repairing intestinal neural circuits with congenital deficits or damaged by injury or disease.

The ENS encompasses the intrinsic neuroglia networks of the gastrointestinal (GI) tract that are essential for digestive function and gut homeostasis (*Furness, 2006*). In vertebrates, assembly of the ENS begins during embryogenesis with invasion of the foregut by a small founder population of

neural crest (NC) cells that proliferate and colonise the entire GI tract, generating diverse types of enteric neurons and glial cells organised into networks of interconnected ganglia (*Heanue and Pachnis, 2007*). ENS development depends on the integrated activity of NC cell lineage-intrinsic programmes and signals from surrounding non-neuroectodermal gut tissues, which ultimately determine the organisation and physiological properties of intestinal neuroglial networks (*Avetisyan et al., 2015*; *Rao and Gershon, 2018*). Despite considerable progress in understanding the developmental mechanisms underpinning the assembly of intestinal neural circuits, much less is known about the dynamics of ENS cell lineages in adult animals, during homeostasis or in response to gut pathology. The predominant view holds that the vast majority of enteric neurons in the mammalian ENS are born during embryogenesis and early postnatal stages and remain functionally integrated into the intestinal circuitry throughout life (*Bergner et al., 2014*; *Joseph et al., 2011*; *Laranjeira et al., 2011*; *Pham et al., 1991*). Likewise, enteric glial cells (EGCs) are generally quiescent, with only a small fraction proliferating at any given time (*Joseph et al., 2011*; *Kabouridis et al., 2015*). Despite this static view of the ENS at homeostasis, lineage tracing experiments in mice have provided evidence that under experimental conditions, such as chemical injury of the ganglionic plexus and bacterial infection, a small fraction of Sox10$^+$ and Sox2$^+$ EGCs can differentiate into neurons (*Belkind-Gerson et al., 2017*; *Belkind-Gerson et al., 2015*; *Laranjeira et al., 2011*). However, a recent study has argued that a population of Sox10$^-$Nestin$^+$ ENS cells undergo extensive proliferation and neuronal differentiation even under physiological conditions, replenishing enteric neurons continuously lost to apoptosis (*Kulkarni et al., 2017*). Although fundamental tenets of this proposition are not supported by available experimental evidence (*Joseph et al., 2011*; *Laranjeira et al., 2011*; *White et al., 2018*), it highlights critical but unresolved questions regarding the cellular and molecular mechanisms underpinning the maintenance and regenerative potential of the ENS in vertebrates.

To address these questions, we investigated the ENS of zebrafish, an excellent model organism for studies on NSCs and neural regeneration in vertebrates. Using genetic lineage tracing, gene expression profiling, correlative light and electron microscopy (CLEM), live imaging, and computational modelling, we demonstrate that the non-neuronal compartment of the zebrafish ENS expresses the transgenic reporter *Tg(her4.3:EGFP)* and shares properties with mammalian EGCs and brain NSCs. *Tg(her4.3:EGFP)*$^+$ ENS cells exhibit morphological features and express genes characteristic of mammalian enteric glia, but canonical glial markers are undetectable. More akin to functional properties of radial glial cells (RGCs) of the zebrafish brain, EGFP$^+$ ENS cells proliferate and undergo constitutive neuronal differentiation which is under the control of Notch signalling. Together, our studies demonstrate the *in vivo* neurogenic potential of enteric glia in vertebrates and reveal previously unanticipated similarities to NSCs in the brain.

## Results

### Expression of canonical glial markers is undetectable in the zebrafish ENS

To pave the way for a systematic search for cells harbouring neurogenic potential in the ENS of non-amniotic vertebrates, we first set out to characterise the non-neuronal compartment of the zebrafish ENS, the most likely source of enteric neural progenitors. Initially, we combined the *SAGFF234A* Gal4 transcriptional activator gene trap with the *UAS:GFP* transgene in order to generate *SAGFF234A; UAS:GFP* animals, in which ENS progenitors and enteric neurons were labelled with GFP (*Heanue et al., 2016a*; *Kawakami et al., 2010*). In 7 day post fertilisation (dpf) larvae the majority of GFP$^+$ cells (93.76% ± 2.99) co-expressed the pan-neuronal marker HuC/D (*Figure 1—figure supplement 1A,D*), suggesting that in comparison to mammals, in which EGCs outnumber enteric neurons (*Gabella, 1981*; *Rühl, 2005*), the non-neuronal ENS cell population of zebrafish is considerably smaller. To support this supposition, we also quantified the proportion of neurons within the ENS of *Tg(−4725sox10:Cre;βactin-LoxP-STOP-LoxP-hmgb1-mCherry)* transgenic fish (hereafter abbreviated as *Tg(sox10:Cre;Cherry)*) in which *sox10*-driven Cre recombinase activates a nuclear Cherry reporter in early NC cells and all derivative lineages, including the ENS (*Rodrigues et al., 2012*; *Wang et al., 2011b*). Although less efficient than a previously published sox10Cre/reporter combination, *Tg(sox10:Cre;ef1a:loxP-GFP-loxP-DsRed2)* (*Rodrigues et al., 2012*; *Figure 1—figure supplement*

*1C*), *Tg(sox10:Cre;Cherry)* labels equivalent proportions of neurons and non-neuronal cells (*Figure 1—figure supplement 1D*). Consistent with the analysis of *SAGFF234A;UAS:GFP* animals, the majority of Cherry+ cells (84.79 ± 7.70%) in the gut of 7 dpf *Tg(sox10:Cre;Cherry)* larvae were positive for HuC/D (*Figure 1A,C*). Similar analysis in adult (≥3 months old) *Tg(sox10:Cre;Cherry)* zebrafish showed that, although the fraction of non-neuronal Cherry+ cells was higher relative to 7 dpf larvae, even at this stage the majority of ENS+ cells (65.49 ± 4.8%) were neurons (*Figure 1B,C*). Therefore, the non-neuronal compartment in the zebrafish ENS is notably smaller relative to its mammalian counterpart.

All non-neuronal cells of the mammalian ENS are identified as enteric glia expressing combinations of the canonical glial markers S100β, GFAP and BFABP (*Hao et al., 2016*; *Young et al., 2003*). To determine whether these marker proteins are also expressed in the zebrafish ENS, we used antibodies raised against them to immunostain 7 dpf larvae, a stage when organised intestinal motility patterns controlled by gut-intrinsic neural networks are clearly evident (*Heanue et al., 2016a*; *Holmberg et al., 2007*; *Kuhlman and Eisen, 2007*). Surprisingly, no signal was detected in the ENS of zebrafish at this stage (*Figure 1D* and *Figure 1—figure supplement 1E–F*). Immunostaining signal detected with two antibodies specific for zebrafish GFAP (*Baker et al., 2019*; *Trevarrow et al., 1990*) was likely to represent cross-reactivity with non-neuroectodermal gut tissues, as it persisted in *ret* mutant larvae, which lack enteric neuroglia networks (*Figure 1—figure supplement 1G–J*; *Heanue et al., 2016a*). Immunostaining signal for GFAP has previously been reported in the ENS (*Baker et al., 2019*; *Kelsh and Eisen, 2000*), however in our experiments the expression is not apparently within the NC-derived lineages. Consistent with the immunostaining, expression of the *Tg(gfap:GFP)* transgene (*Bernardos and Raymond, 2006*) was also undetectable in the gut of 7 dpf larvae (*Figure 1E*). In contrast to the ENS, these immunostaining and transgenic reagents identified the expected signal in the spinal cord (*Figure 1—figure supplement 1K–P*). To ascertain that the lack of glia marker expression was not due to delayed maturation of enteric glia, we also immunostained adult zebrafish gut for GFAP, S100β, BFABP and (in the case of *gfap:GFP* transgenics) GFP. Similar to 7 dpf animals, no apparent ENS-specific expression of these markers or the *gfap:GFP* transgene was detected in the adult gut (*Figure 1F–G*, *Figure 1—figure supplement 1Q–R*). Finally, contrary to reports indicating expression of Nestin in non-neuronal cells of mammalian enteric ganglia (*Kulkarni et al., 2017*), no expression of the *nestin:GFP* transgene was detected in the ENS of adult zebrafish (*Figure 1—figure supplement 1S*). Taken together, our studies demonstrate that the non-neuronal compartment of the zebrafish ENS is considerably smaller relative to its mammalian counterpart and cannot be labelled by immunohistochemical reagents commonly used for the identification of enteric glia.

## Non-neuronal cells of the zebrafish ENS share with mammalian EGCs early NC cell and ENS progenitor markers

To explore further the gene expression profile of the non-neuronal ENS cell compartment in zebrafish, we carried out bulk RNA sequencing of fluorescent-labelled nuclei (nRNAseq) isolated from *Tg(sox10:Cre;Cherry)* adult gut muscularis externa. This strategy, which we described recently (*Obata et al., 2020*), avoids lengthy protocols of tissue dissociation and cell isolation that are often associated with considerable cell damage. Since the available transgenic tools did not allow us to label specifically the non-neuronal ENS cell compartment, bulk nRNAseq was performed on nuclei purified by FACS (fluorescent-activated cell sorting) representing both the Cherry+ (entire ENS) and Cherry- (non-ENS) muscularis externa cell populations of *Tg(sox10:Cre;Cherry)* zebrafish gut (*Figure 2A* and *Figure 2—figure supplement 1A*; see also Materials and Methods). Principal component analysis (PCA) demonstrated a clear separation of the Cherry+ and Cherry- nuclear transcriptomes along PC1 (*Figure 2—figure supplement 1B*), indicating that variability along this axis is determined predominantly by the lineage origin (NC vs non-NC) of the two cell populations. As expected, genes associated with non-NC tissues, such as smooth muscle cells (*myh11a*, *cald1a*, *srfa*, *gata6*), interstitial cells of Cajal (*ano1*, *kita*, *kitb*) and immune cells (*lcp1*, *lck*, *lyz*), were upregulated in the Cherry- nuclear transcriptome (*Figure 2B*). Conversely, genes associated with the NC-derived ENS lineages (such as *elavl3*, *elavl4*, *ret*, *vip*, *chata*, *sox10*) were upregulated in the Cherry+ nuclear population (*Figure 2B*) (GEO database GSE145885; *Supplementary file 1*; an interactive data viewer to explore the analysed data can be found here: https://biologic.crick.ac.uk/ENS). Furthermore, gene ontology (GO) terms enriched in the Cherry+ nuclear population were associated with nervous system development and function (*Figure 2—figure supplement 1C–E*). Finally, direct comparison of the Cherry+ dataset to

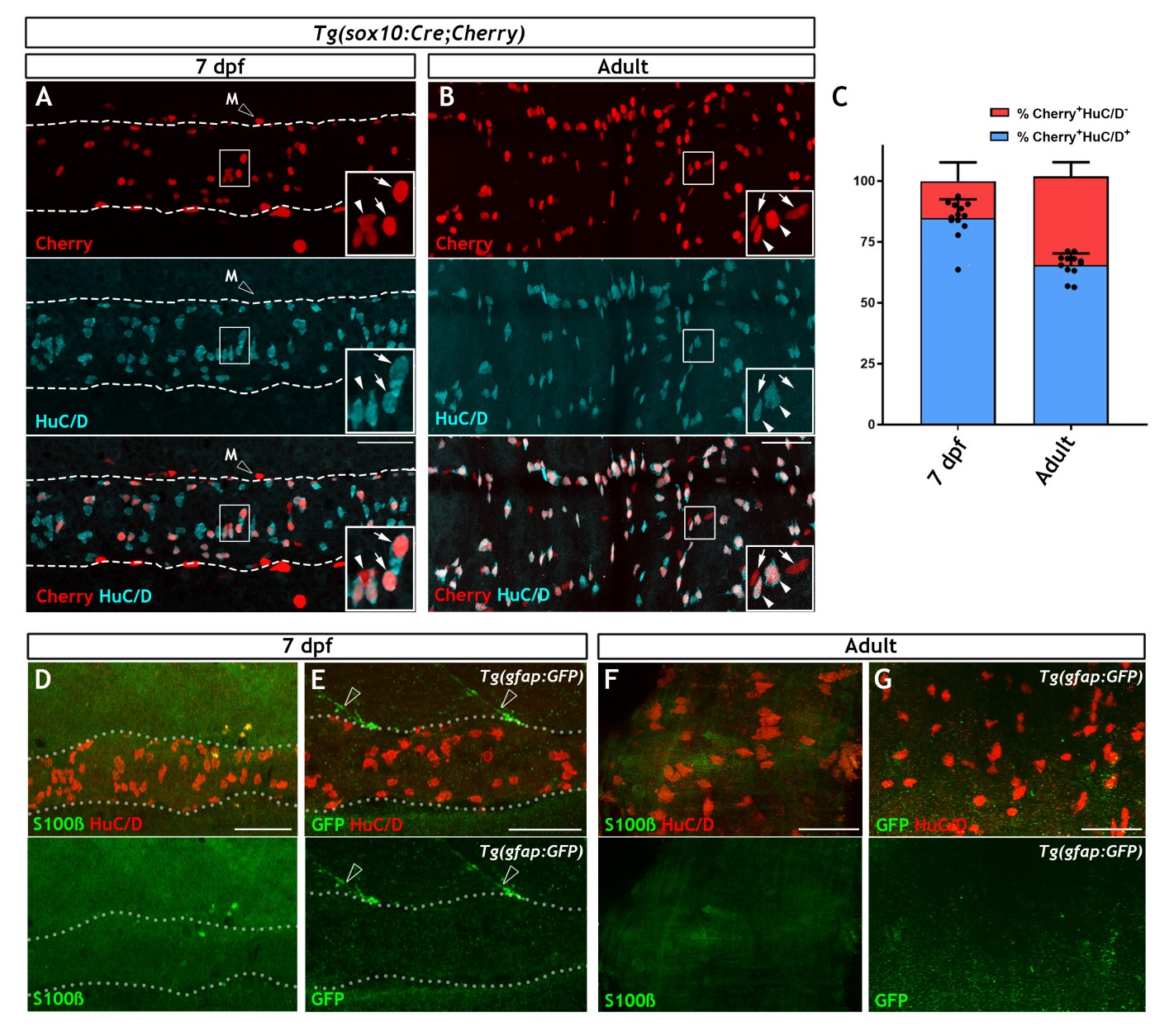

**Figure 1.** The non-neuronal compartment of the zebrafish ENS is relatively small and is not identified using canonical glial markers. (A) Confocal images of the gut of 7 dpf *Tg(sox10:Cre;Cherry)* larvae immunostained for Cherry (red, top) and HuC/D (cyan, middle) (n = 13). The bottom panel is a merge of the Cherry and HuC/D signals. Inset shows a high magnification of the boxed area. Arrows point to Cherry+HuC/D+ cells and an arrowhead points to a Cherry+HuC/D- cell. Dotted line delineates the gut. Open arrowhead indicates a Cherry+ NC-derived melanocyte (M), which is present outside the intestine. (B) Confocal images of the ENS in adult zebrafish intestine immunostained for Cherry (red, top) and HuC/D (cyan, middle) (n = 13). The bottom panel is a merge of the Cherry and HuC/D signals. Inset shows a high magnification of the boxed area. Arrowheads point to Cherry+HuC/D+ cells and arrows point to Cherry+HuC/D- cells. (C) Quantification of the neuronal (Cherry+HuC/D+) and non-neuronal (Cherry+HuC/D-) cellular compartments within the *sox10*-lineage at 7 dpf and adult zebrafish, n = 13 biological replicates, data are given as mean ± SD. (D) Confocal images of the gut of 7 dpf zebrafish larvae immunostained for S100β (green) and HuC/D (red). No S100β signal was detected in the ENS, despite abundant neurons throughout the intestine (n = 30). (E) Confocal images of the gut of 7 dpf *Tg(gfap:GFP)* larvae immunostained for GFP (green) and HuC/D (red). No GFP signal was visible within the intestine despite abundant HuC/D+ neurons (n = 50). GFP+ fibres associated with spinal nerves are observed descending towards the gut but never enter the intestine (open arrowheads). Dotted lines in D and E delineate the gut. (F) Immunostaining of the ENS of adult zebrafish with S100β (green) and HuC/D (red) (n = 5). (G) Immunostaining of the ENS of adult *Tg(gfap:GFP)* zebrafish with GFP (green) and HuC/D (red) (n = 13). S100β (F) and GFP (G) signal was absent despite the presence of HuC/D+ neurons. All confocal images are max projections of short confocal stacks. 50 μm scale bars shown in merge panels.

*Figure 1 continued on next page*

eLife Research article

Developmental Biology | Neuroscience

the transcriptional profile of enteric neurons from 7 dpf larvae expressing the *Tg(phox2b:EGFP)*$^{w37}$ transgene (*Roy-Carson et al., 2017*), identified a large cohort of shared genes (including *phox2bb*, *ret*, *elavl3*, *elavl4*, *vip*, *nmu*) that presumably reflect the neural component of the mixed Cherry$^+$ nuclear population (*Figure 2C*, yellow dots, *Figure 2—figure supplement 1F* and *Supplementary file 2*).

**Figure supplement 1.** ENS lineage tracing shows that there is a small non-neuronal lineage that is not detectable using antibodies for the canonical glial markers BFABP, GFAP nor with transgenic reporters.

To identify genes expressed by the non-neuronal compartment of the zebrafish ENS, we next compared the Cherry$^+$ dataset to a recently reported transcriptome of mouse EGCs, which presented a list of the 25 most highly expressed genes in PLP1$^+$ enteric glia (*Rao et al., 2015*). Zebrafish orthologues for several genes in this list were enriched in the Cherry$^+$ transcriptome (*Figure 2—figure supplement 1G*), suggesting that they are expressed by the non-neuronal cells of the zebrafish ENS. Among these genes were *sox10*, *foxd3* and *plp1*, which in mammals are expressed by early NC cells and ENS progenitors and maintained in enteric glia (*Dyachuk et al., 2014*; *Hari et al., 2012*; *Mundell and Labosky, 2011*; *Mundell et al., 2012*; *Weider and Wegner, 2017*), as well as genes with established association to glial cells, such as *col28a1* (*Grimal et al., 2010*), *ptprz1a* and *ptprz1b* (*Fujikawa et al., 2017*). In a similar strategy, we have also compared the Cherry$^+$ datatset to a single cell transcriptomic dataset of mouse ENS neurons and glia (*Zeisel et al., 2018*). We have identified the genes from this mouse dataset that are differentially expressed between mouse ENS glia and neurons and determined their zebrafish orthologues (*Supplementary file 3*). We show that 366 mouse ENS neuron-enriched genes have orthologues present in our zebrafish Cherry$^+$ transcriptome dataset, including *elavl3*, *elavl4*, *prph*, and *phox2bb*, and likely reflect the neuronal component of our bulk dataset (*Figure 2—figure supplement 2A,B*, *Supplementary file 4*). We also show that 63 mouse ENS glia-enriched genes have orthologues present in the zebrafish Cherry$^+$ dataset, suggesting that these glial expressed genes are detected in the non-neuronal component of the zebrafish ENS (*Figure 2—figure supplement 2A,C*, *Supplementary file 5*), including *sox10*, *foxd3*, *plp1b*, and the additional neural crest marker *zeb2b* (*Delalande et al., 2008*), *sox2*, which is expressed by mouse ENS progenitors and adult EGCs (*Belkind-Gerson et al., 2017*; *Heanue and Pachnis, 2011*), and the CNS glia associated gene *vim* (*Deng et al., 2013*). Significantly, we do not observe canonical glial markers *gfap*, *s100b* and *fabp7a/b* amongst these genes, consistent with the failure to detect expression of these markers by immunostaining analysis (*Figure 1F,G* and *Figure 1—figure supplement 1Q,R*), though we cannot exclude the possibility that such markers may be revealed by in depth sequencing of single cells. In a final strategy to identify genes associated with the non-neuronal component of the zebrafish ENS, we applied a strategy that was not reliant on cross-species comparisons. Having delineated the neural component of the Cherry$^+$ transcriptome (*Figure 2C*, yellow dots, and *Figure 2—figure supplement 1F*), we removed this cohort of genes in order to enrich for transcripts of the non-neuronal ENS cell compartment (*Figure 2C*, black dots, *Figure 2—figure supplement 1H* and Suppl. File 6). This strategy highlighted several genes that were identified by our previous analysis, including *sox10* and *foxd3*. Numerous additional genes were identified, including *tfap2a*, a gene required in early NC cells (*Knight et al., 2003*; *Wang et al., 2011a*), and *sox2*. Expression of *sox10*, *foxd3* and *sox2* in the non-neuronal compartment of the zebrafish ENS was validated by combining multiplex fluorescence *in situ* hybridisation (RNAscope) with immunostaining for HuC/D and the Cherry reporter on muscularis externa preparations from the gut of adult *Tg(sox10:Cre;Cherry)* zebrafish (*Figure 2D–F*). Together, these experiments indicate that, despite our failure to detect expression of commonly used EGC markers, the transcriptomes of the non-neuronal compartment of the zebrafish ENS and mammalian enteric glia have considerable overlap, including genes associated with early NC cells and ENS progenitors.

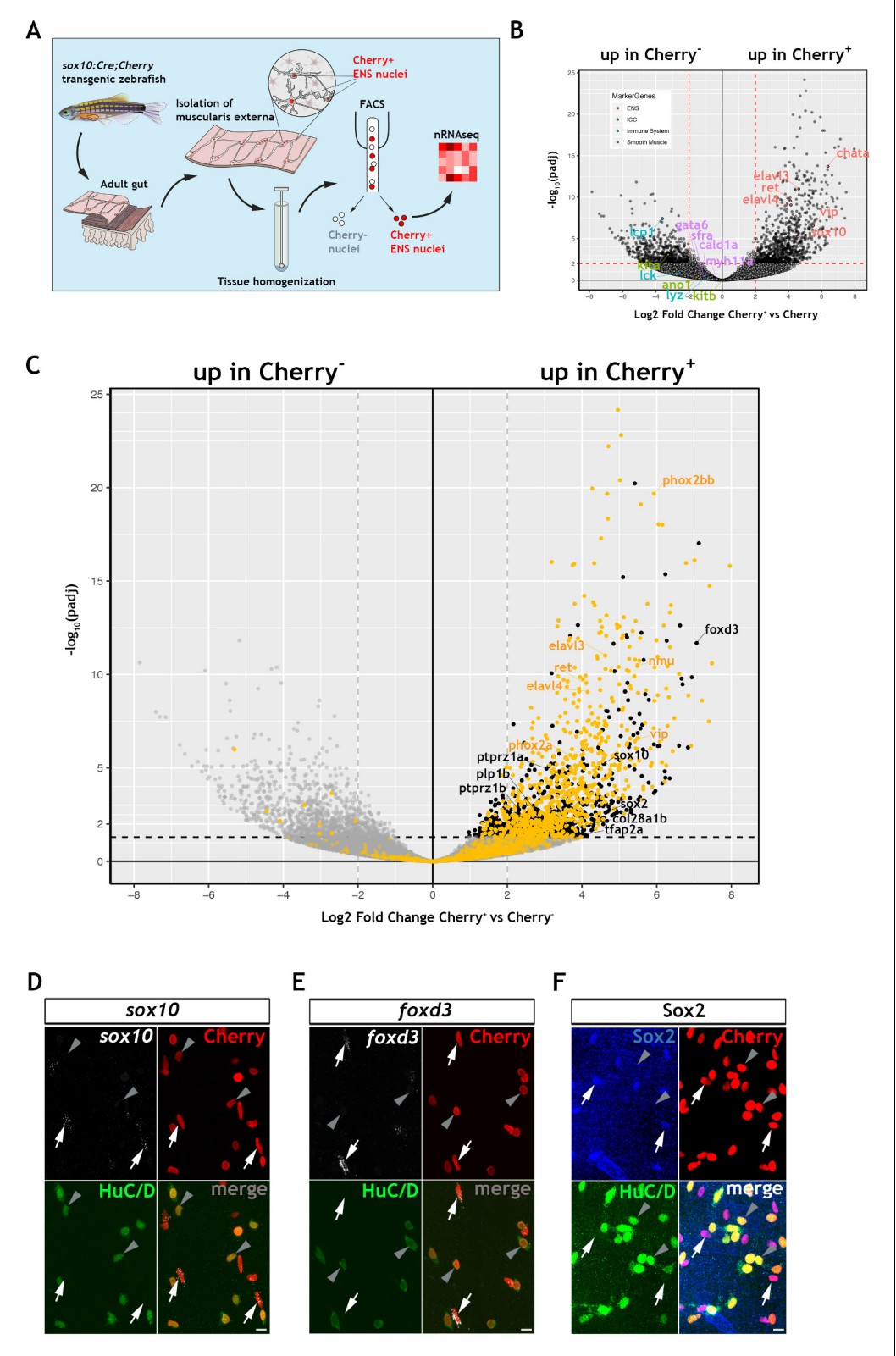

**Figure 2.** Transcriptomic profiling of the adult zebrafish ENS. (**A**) Experimental strategy for the isolation of ENS nuclei from adult *Tg(sox10:Cre;Cherry)* guts and nuclear RNAseq. Five biological replicates were performed per condition. (**B**) Volcano plot shows mean log$_2$ fold-change (x axis) and significance (-log$_{10}$ adjusted p-value) (y axis) of genes differentially expressed in Cherry$^+$ relative to Cherry$^-$ nuclei. Genes characteristic of the ENS are highlighted in red and are more abundant in Cherry$^+$ nuclei, whereas genes characteristic of non-neuroectodermal lineages, such as smooth muscle

*Figure 2 continued on next page*

*Figure 2 continued*

(purple), interstitial cells of Cajal (green) and immune associated (blue), are more abundant in Cherry⁻ nuclei. (C) Volcano plot (as in B) in which genes previously identified in a transcriptional characterization of larval ENS neurons (**Roy-Carson et al., 2017**) are shown in yellow. These include established neuronal markers, such as *phox2bb, ret, elavl3, elavl4, vip*, and *nmu*. Genes enriched in the Cherry⁺ nuclear population but absent from the larval ENS neuron transcriptome are shown in black. These include *sox10, foxd3, sox2, plp1*, the mammalian orthologues of which are expressed by mouse EGCs, *tfap2a*, a gene required for early NC development, *col28a1b*, whose mammalian orthologue is a peripheral glial marker, as well as *ptprz1a*, and *ptprz1b*, which have been identified in glioblastoma stem cells. Genes with padj <0.05 (Log$_{10}$p-value<1.3) and/or log$_2$FC < 0 are shown in grey. (D,E) Confocal images of fluorescent *in situ* hybridization (RNAscope) using probes for *sox10* (D) and *foxd3* (E) on adult *Tg(sox10:Cre;Cherry)* gut muscularis externa preparations immunostained for Cherry (ENS lineage) and HuC/D (ENS neurons). Signal for both *sox10* and *foxd3* (white arrows) corresponds to non-neuronal cells (Cherry⁺HuC/D⁻, arrows) but was absent from enteric neurons (Cherry⁺HuC/D⁺, arrowheads). (F) Immunostaining of adult *Tg(sox10:Cre;Cherry)* gut for Sox2 (blue), Cherry (red) and HuC/D (green). Sox2 is expressed specifically by non-neuronal ENS cells. Biological replicates: D, n = 4; E, n = 6; F, n = 5. All confocal images are max projections of short confocal stacks. 10 µm scale bars shown in merge panels.

The online version of this article includes the following figure supplement(s) for figure 2:

**Figure supplement 1.** Transcriptional profiling of adult zebrafish ENS nuclei identifies profiles indicative of both neurons and glia.
**Figure supplement 2.** Comparison of the zebrafish ENS transcriptome to a single cell transcriptomic dataset of mouse ENS neurons and ENS glia.
**Figure supplement 3.** Interrogation of a mouse single cell transcriptomic dataset to identify genes characterising mouse ENS neurons and ENS glia.

## Non-neuronal cells in the adult zebrafish ENS express the Notch activity reporter *Tg(her4.3:EGFP)*

In mammals, Notch signalling promotes enteric gliogenesis by attenuating a cell-autonomous neurogenic programme of ENS progenitors (**Okamura and Saga, 2008**), but the expression of Notch target genes in adult EGCs is unclear. Moreover, the transgenic Notch activity reporter *Tg(her4.3:EGFP)* (see Materials and Methods for the nomenclature of this transgene) marks NSCs and neural progenitors in the zebrafish brain (**Alunni and Bally-Cuif, 2016**; **Yeo et al., 2007**). Given the fact that the non-neuronal component of the ENS appears to be enriched for progenitor markers, and our desire to find a suitable transgenic tool to facilitate further study, we have examined whether the Notch activity reporter *Tg(her4.3:EGFP)* is also observed in non-neuronal cells of the zebrafish ENS. We examined the adult gut for expression of *Tg(her4.3:EGFP)* expressing cells. This analysis identified a network of GFP⁺ cells in the muscularis externa of the gut that was closely associated with enteric neurons and their projections (**Figure 3A** and **Figure 3—figure supplement 1A**). To provide direct evidence that *Tg(her4.3:EGFP)* expressing cells are integral to the ENS, we introduced the *her4.3:EGFP* transgene into the *Tg(sox10:Cre;Cherry)* genetic background and immunostained gut preparations from adult *Tg(her4.3:EGFP;sox10:Cre;Cherry)* animals for GFP, HuC/D and Cherry. As expected, GFP⁺ cells were negative for HuC/D but expressed the Cherry reporter (**Figure 3C**), indicating that they belong to the non-neuronal compartment of the ENS. Consistent with this idea, GFP⁺ cells co-expressed *sox2* and *sox10* (**Figure 3D,E**), which were identified by our transcriptomic analysis as genes expressed by the non-neuronal compartment of the zebrafish ENS. We observed heterogeneity within the non-neuronal component: whereas Sox2 was widely expressed in the GFP⁺HuC/D⁻ cell population in *Tg(her4.3:EGFP;sox10:Cre;Cherry)* tissue, *foxd3* was expressed in only a proportion of the GFP⁺HuC/D⁻/Sox2⁺ cells (**Figure 3F**). The GFP⁺HuC/D⁻ cell population in *Tg(her4.3:EGFP;sox10:Cre;Cherry)* represented approximately a quarter (24.20 ± 5.18%) of all Cherry⁺ ENS cells, but 12.93 ± 5.33% of Cherry⁺ cells were negative for both GFP and HuC/D (Cherry⁺GFP⁻HuC/D⁻) (**Figure 3B**). Therefore, the majority of non-neuronal ENS cells in adult zebrafish gut can be identified by the expression of the Notch activity reporter *Tg(her4.3:EGFP)*.

## GFP⁺ cells in the ENS of adult *Tg(her4.3:EGFP)* zebrafish have morphological characteristics of mammalian EGCs

To provide evidence that *Tg(her4.3:EGFP)* expressing cells in the zebrafish ENS are equivalent to mammalian EGCs, we characterised the morphology of GFP⁺ cells in the gut of *Tg(her4.3:EGFP)* transgenics. At the light microscopy level GFP⁺ cells were highly branched and fell into four morphological groups that generally corresponded to the four morphological subtypes of mouse EGCs (Types I-IV) (**Figure 3—figure supplement 1B–E**; **Boesmans et al., 2015**; **Gulbransen and Sharkey, 2012**). Although the zebrafish ENS lacks distinct ganglia found in mammalian systems, the presence

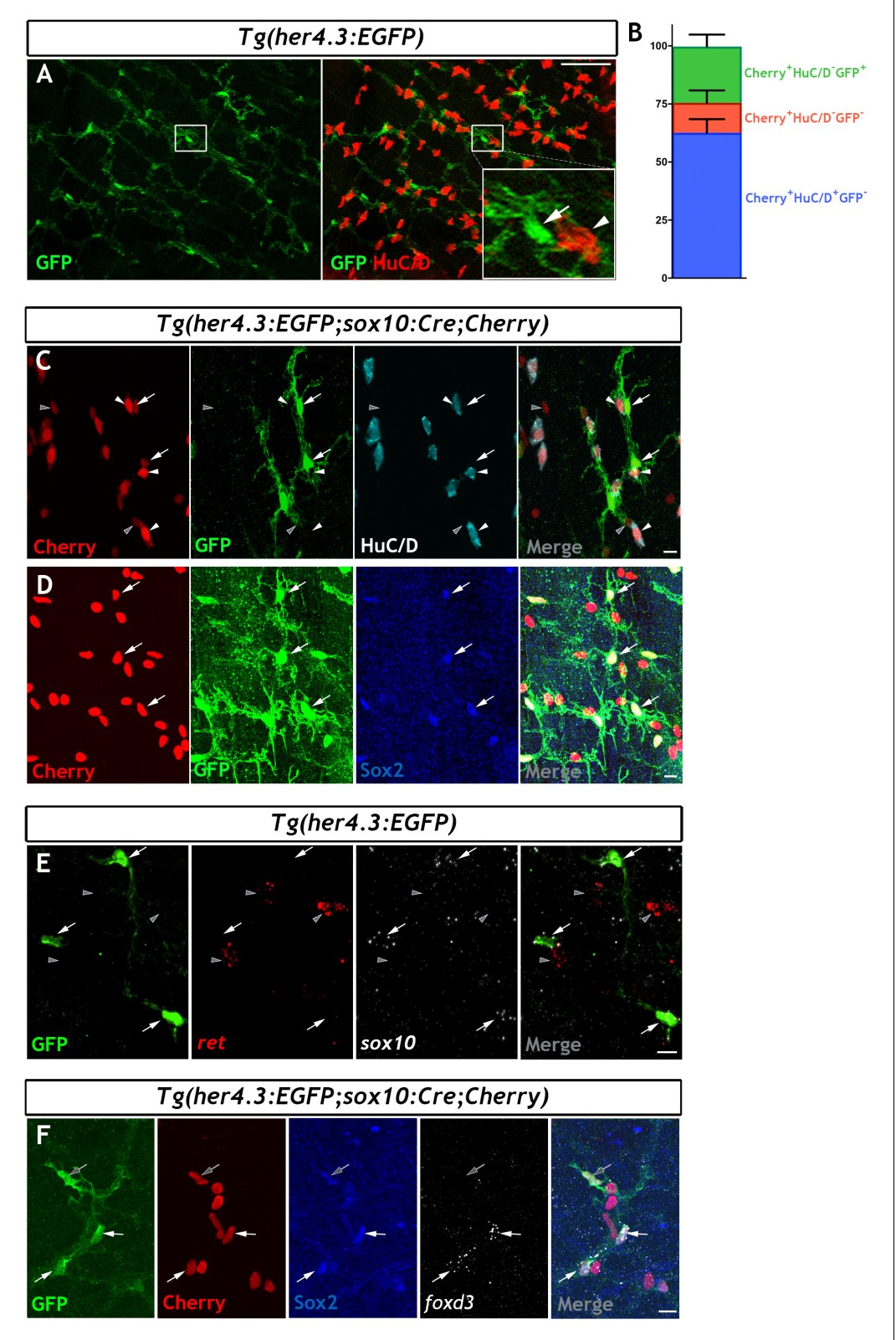

**Figure 3.** The *her4.3:EGFP* transgene is a novel marker of the non-neuronal cell population in the adult zebrafish ENS. (**A**) Confocal images of adult *Tg(her4.3:EGFP)* zebrafish gut immunostained for GFP (green) and HuC/D (red). Inset is a high magnification of boxed area showing that GFP$^+$ cells (arrow) are closely associated with HuC/D$^+$ neurons (arrowhead) (n = 70). (**B**) Quantification of neuronal (Cherry$^+$ HuC/D$^+$GFP$^-$, blue) and non-neuronal cell populations (Cherry$^+$HuC/D$^-$GFP$^+$ and Cherry$^+$HuC/D$^-$GFP$^-$, green and red, respectively) in the ENS of adult *Tg(her4.3:EGFP;sox10:Cre;Cherry)*
*Figure 3 continued on next page*

Figure 3 continued

zebrafish (n = 3). Data are given as mean ± SD. (C) Confocal images of the ENS from adult *Tg(her4.3:EGFP;sox10:Cre;Cherry)* zebrafish immunostained for Cherry (red), GFP (green) and HuC/D (cyan). Note the presence of Cherry⁺HuC/D⁻GFP⁺ (arrows) and Cherry⁺ HuC/D⁻ GFP⁻ (grey arrowheads) cells as well as the presence of Cherry⁺HuC/D⁺GFP⁻ neurons (white arrowheads) (n = 3). Note that Cherry⁺ nuclei are of equivalent size in Cherry⁺HuC/D⁻GFP⁺ (arrows), Cherry⁺ HuC/D⁻ GFP⁻ (grey arrowheads) cells, and Cherry⁺HuC/D⁺GFP⁻ neurons (white arrowheads). (D) Immunostaining of adult *Tg(her4.3:EGFP;sox10:Cre;Cherry)* gut with antibodies for Cherry (red), GFP (green) and Sox2 (blue). Arrows point to cells expressing all three markers (n = 3). (E) RNAscope analysis for *ret* (red) and *sox10* (white) on ENS preparations from adult *Tg(her4.3:EGFP)* zebrafish guts immunostained for GFP (green). Note that GFP⁺ cells (arrows) express *sox10* and are found in close proximity to *ret*⁺GFP⁻ enteric neurons (grey arrowheads) (n = 4). (F) Combined RNA scope for *foxd3* and immunostaining for GFP, Cherry and Sox2 on adult *Tg(her4.3:EGFP;sox10:Cre;Cherry)* gut shows that *foxd3* and Sox2 are co-expressed in some ENS cells (white arrows), other GFP⁺ cells express only Sox2 (grey arrows). All confocal images are max projections of short confocal stacks. Scale bars in merge panels: (A) 50 µm (C–E) 10 µm.

The online version of this article includes the following figure supplement(s) for figure 3:

**Figure supplement 1.** Her4.3GFP transgenic line identifies cells with morphologies indicative of distinct subtypes of EGCs in the adult ENS.

of GFP⁺ cells in the myenteric layer, in close association with HuC/D⁺ cells, with multiple processes wrapping around the HuC/D⁺ cell bodies, is reminiscent of Type I mammalian EGCs (*Figure 3—figure supplement 1B*; *Hanani and Reichenbach, 1994*; *Boesmans et al., 2015*). Moreover, elongated GFP⁺ cells in the myenteric layer with cell bodies and processes that follow along AcTu⁺ neuronal cell processes show clear parallels with Type II mammalian EGCs (*Figure 3—figure supplement 1C*; *Hanani and Reichenbach, 1994*; *Boesmans et al., 2015*). GFP⁺ cells were also found within the mucosa in close proximity to the intestinal epithelium (*Figure 3—figure supplement 1D*), similar to Type III mucosal EGCs located within the lamina propria of the mammalian gut (*Boesmans et al., 2015*; *Kabouridis et al., 2015*). And finally, bipolar GFP⁺ cells found within the smooth muscle layers and associated with AcTu⁺ neuronal fibres are reminiscent of Type IV mammalian glia (*Figure 3—figure supplement 1E*; *Boesmans et al., 2015*).

Mammalian EGCs have unique ultrastructural features and establish characteristic contacts with enteric neurons and their projections (*Gabella, 1972*; *Gabella, 1981*). To determine whether similar features are exhibited by the GFP⁺ ENS cell population in *Tg(her4.3:EGFP)* zebrafish, we analysed EGFP⁺ cells in *Tg(her4.3:EGFP;SAGFF217B;UAS:mmCherry)* transgenics using CLEM (*Müller-Reichert and Verkade, 2012*). In these animals, EGFP marks non-neuronal ENS cells while Cherry, which is driven by the binary reporter *Tg(SAGFF217B;UAS:mmCherry)* (*Kawakami et al., 2010*), labels a subset of enteric neurons (*Figure 4—figure supplement 1A*). CLEM confirmed the close association of EGFP⁺ cells with enteric neurons and their projections (*Figure 4*, *Figure 4—figure supplement 1B,C* and *Video 1*). Processes emanating from EGFP⁺ cells could extend to 18 µm and directly contacted enteric neurons (*Figure 4B,D* and *Figure 4—figure supplement 1C*), but similar to mammalian EGCs (*Gabella, 1981*) they did not form complete 'capsules' around neuronal somata, allowing large parts of enteric neurons to be in direct contact with adjacent cells (*Figure 4A,B*, *Figure 4—figure supplement 1C* and *Video 1*). EGFP⁺ cells also extended complex sheet-like extensions, which frequently enclosed and/or subdivided the tightly packed bundles of neural projections into sectors (*Figure 4D*, *Figure 4—figure supplement 1C* and *Video 1*). Scarcity of cytoplasm and deep nuclear crenations, characteristic features of mammalian EGCs and other populations of peripheral glial cells (*Gabella, 1981*), were also found in the nuclei of EGFP⁺ cells (*Figure 4B,D* and *Figure 4—figure supplement 1C*).

Taken together we show that like mammalian EGCs, the *her4.3:EGFP*⁺ population share a lineage with ENS neurons, are found within the myenteric layer in close association with ENS neurons, have distinctive morphologies reminiscent of the four types of mammalian EGCs, have ultrastructural features of mammalian ENS glia, and express multiple well-established mammalian EGC markers. This weight of evidence leads us to conclude that *her4.3:EGFP*⁺ cells constitute the zebrafish EGC population, and therefore now define them as such. Henceforth, we will be referring to *Tg(her4.3:EGFP)* expressing cells in the adult zebrafish ENS as EGCs.

## Developmental profile of zebrafish EGCs

To examine the developmental profile of zebrafish EGCs, we immunostained *Tg(her4.3:EGFP; SAGFF234A;UAS:mmCherry)* transgenics for GFP and Cherry at different developmental stages. At 54 hr post fertilisation (hpf), a stage at which NC cell-derived Cherry⁺ cells are restricted to two distinct

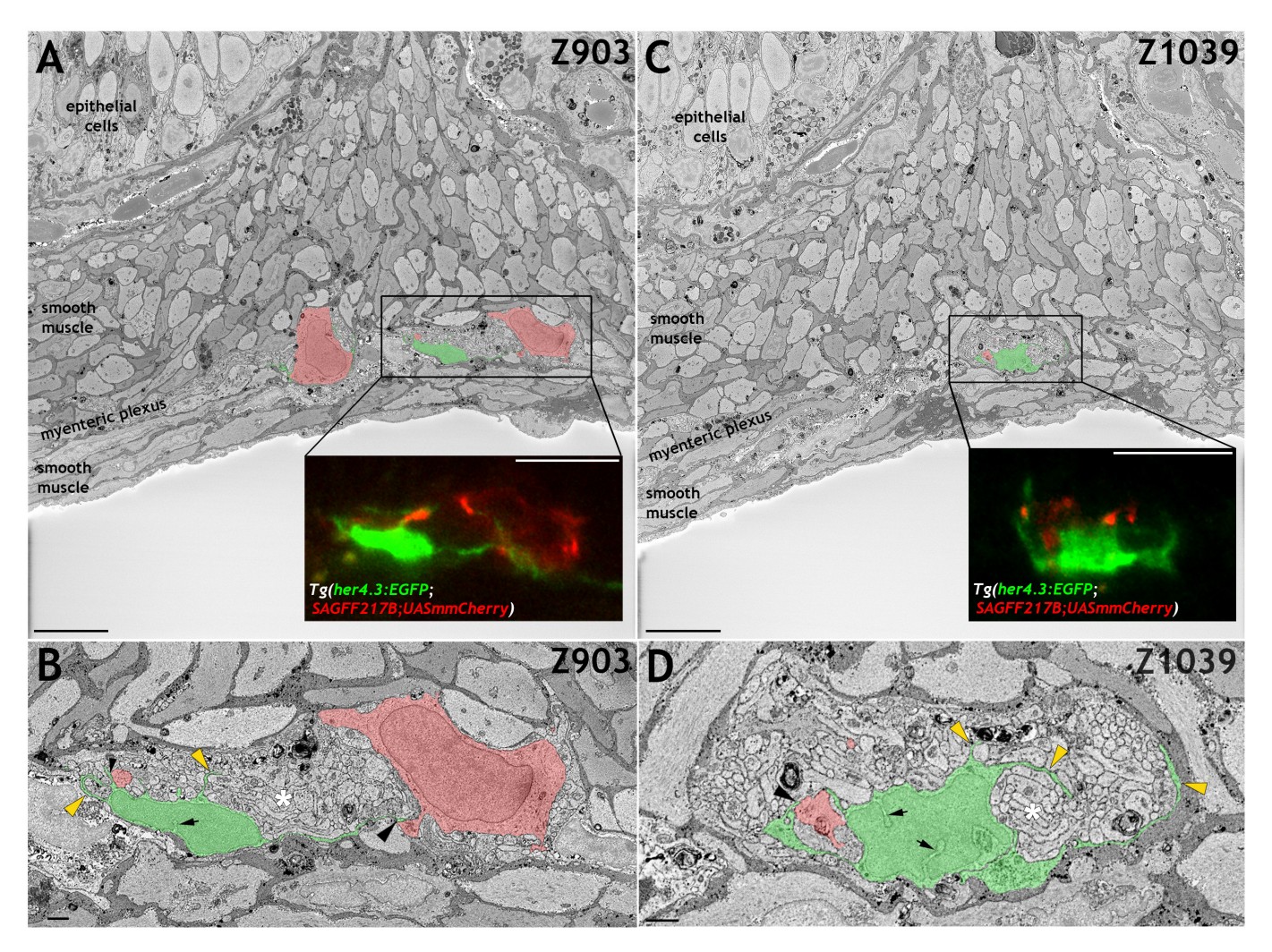

**Figure 4.** *her4.3:EGFP* expressing cells in the adult zebrafish ENS share with mammalian enteric glia characteristic ultrastructural features. (**A and C**) Electron micrographs (z-stack # 903 in **A** and #1039 in **C**) from a 3D region of interest from the midgut of adult *Tg(her4.3:EGFP;SAGFF217;UAS: mmCherry)* zebrafish. Insets shows super-resolution light microscopy images of EGFP⁺ non-neuronal cells and mmCherry⁺ neurons that correspond to the boxed areas of the electron micrograph. The EGFP⁺ cells have a cell soma size of ~79.6 µm³ (**A**) and ~79.1 µm³ (**C**) with projection lengths that range from sheet-like processes of 4 µm to longer extensions of up to 18 µm. For comparison, the mmCherry⁺ neurons have cell soma size of ~398.8 µm³ (**A**, left) and 229.7 µm³ (**A**, right) with projection lengths that range from 16 µm to 55 µm. (**B and D**) High-resolution images of the boxed areas shown in A (**B**) and C (**D**). The EGFP⁺ cells are pseudocoloured in green and enteric neurons in red. Black arrowheads indicate points of contact between EGFP⁺ processes and mmCherry⁺ neurons. Yellow arrowheads indicate GFP⁺ sheet-like extensions that compartmentalise axon bundles (white asterisks). Nuclear crenelations in nuclei of EGFP⁺ cells are indicated with black arrows. Representative images of six regions of interest scanned from two adults. All images are a single z plane. Scale bars: 10 µm (**A, C** and insets **A,C**) and 1 µm (**B,D**).

The online version of this article includes the following figure supplement(s) for figure 4:

**Figure supplement 1.** Correlative light-electron microscopy identifies glial like features of adult *Tg(her4.3:EGFP)* expressing cells.

migratory columns along the gut (*Heanue et al., 2016a*), no double positive (Cherry⁺GFP⁺) cells were identified (*Figure 5—figure supplement 1A*). However, at 60 hpf a small number of GFP⁺ cells were discernible within the Cherry⁺ streams of NC cells (*Figure 5—figure supplement 1B*) and became more abundant in 4 dpf larvae (*Figure 5—figure supplement 1C*). To further examine the developmental dynamics of the GFP⁺ cell lineage, we performed time-lapse confocal microscopy of live *Tg(her4.3: EGFP;SAGFF234A;UAS:mmcherry)* embryos at similar stages. Imaging commenced at 56 hpf with the migratory front of mmCherry⁺ NC cell columns positioned at the rostral end of the field of view (*Heanue et al., 2016a*) and continued for 40 hr (1 image every 10 min). Consistent with the analysis

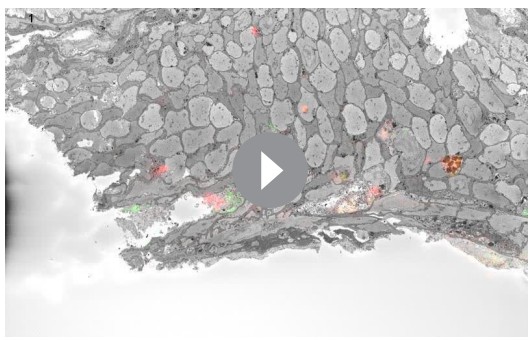

**Video 1.** Correlative light and electron microscopy (CLEM) analysis of the adult *Tg(her4.3:EGFP; SAGFF234A;UASmmCherry)* gut. Mapping of the super-resolution light microscopy volume into the cropped SBF SEM volume using Bigwarp confirmed the identification and localisation of EGFP+ non-neuronal cells and mmCherry+ neurons within a 3D region of interest from the midgut of *Tg(her4.3:EGFP; SAGFF217;UAS:mmCherry)* zebrafish. The EGFP+ cells and mmCherry+ neurons that were false coloured in *Figure 4* and *Figure 4—figure supplement 1* are indicated with green and red arrows, respectively, showing that each forms numerous complex extensions through the volume. Data are shown at 10 frames per second, with 100 nm pixels in XY (cropped to represent a horizontal frame width of 80.5 μ m) and 50 nm pixels in Z (representing a depth of 64.8 μ m).
https://elifesciences.org/articles/56086#video1

performed on fixed embryos, no EGFP+ cells were identified within the mmCherry+ population during the first hours of imaging (*Figure 5A*). However, EGFP+ cells appeared within the columns of mmCherry+ cells at around 62 hpf (*Figure 5B*), more than 90 μm behind the front of migrating mmCherry+ NC cells, and the number of EGFP+ cells increased over the remaining imaging period (*Figure 5C,D*; *Video 2*). On several occasions, we identified individual mmCherry+ cells inducing de novo expression of EGFP (*Video 3*). EGFP+ cells emerged in a rostro-caudal sequence mirroring the wave of ENS neuron maturation (*Heanue et al., 2016b*) but they were almost always located behind the front of migrating enteric NC cells. Relative to the tip of the mmCherry+ migratory column, which was displaced caudally at a constant rate until it reached the caudal end of the FOV, EGFP+ cells on average exhibited minimal rostrocaudal displacement (*Figure 5E*; 132 EGFP+ cells analysed from four fish), suggesting that during ENS development the *her4.3:EGFP* transgene is expressed in post-migratory cells.

Next, we characterised the cell division patterns of the 79 EGFP+ cells that migrated into the field of view or arose de novo during the live imaging period. Of these, 37 cells gave rise to at least one generation of GFP+ progeny. 26 cells (~33%) underwent a single cell division generating two daughters, many of which lost EGFP expression over the course of imaging. In these cases the EGFP expressing cells were not migratory and the EGFP expression diminished and then extinguished. In a proportion of cells (8 cells; ~10%), after a first division event, one or both of the daughter cells underwent a further cell division, generating EGFP+ granddaughters, some of which lost expression of the reporter. For 3 cells (~4%), following two division events, one granddaughter cell underwent a further division to generate a third generation of EGFP+ progeny. Altogether, 53 EGFP+ cells were seen to undergo a cell division event during the imaging period. Therefore, during development *Tg(her4.3:EGFP)* expressing cells are capable of entering the cell cycle but those that do so undergo only a limited number of cell divisions and many of their progeny eventually lose expression of EGFP. Loss of EGFP signal could be associated with neuronal differentiation, since we occasionally identified in the gut of 7 dpf *her4.3:EGFP* transgenic larvae cells that were weakly immunostained for both HuC/D and GFP (*Figure 5—figure supplement 1D*). Taken together, our analysis of *Tg(her4.3:EGFP)* expression during zebrafish development suggests that nascent EGCs are postmigratory NC-derived cells which maintain proliferative and neurogenic potential.

## Proliferation and neuronal differentiation of adult zebrafish EGCs during homeostasis

Enteric glia in adult mammals are generally quiescent with only a small fraction of cells undergoing cell division at any given time (*Joseph et al., 2011*). To examine the proliferative potential of EGCs in adult zebrafish, we immunostained whole-mount gut preparations from adult *Tg(her4.3:EGFP)* transgenics for the proliferation marker mini-chromosome maintenance 5 (MCM5) (*Ryu et al., 2005*). 10.8 ± 4.2% of GFP+ cells were positive for MCM5 (*Figure 6—figure supplement 1*), indicating that in contrast to their mammalian counterparts, a considerable proportion of zebrafish EGCs are cycling during homeostasis.

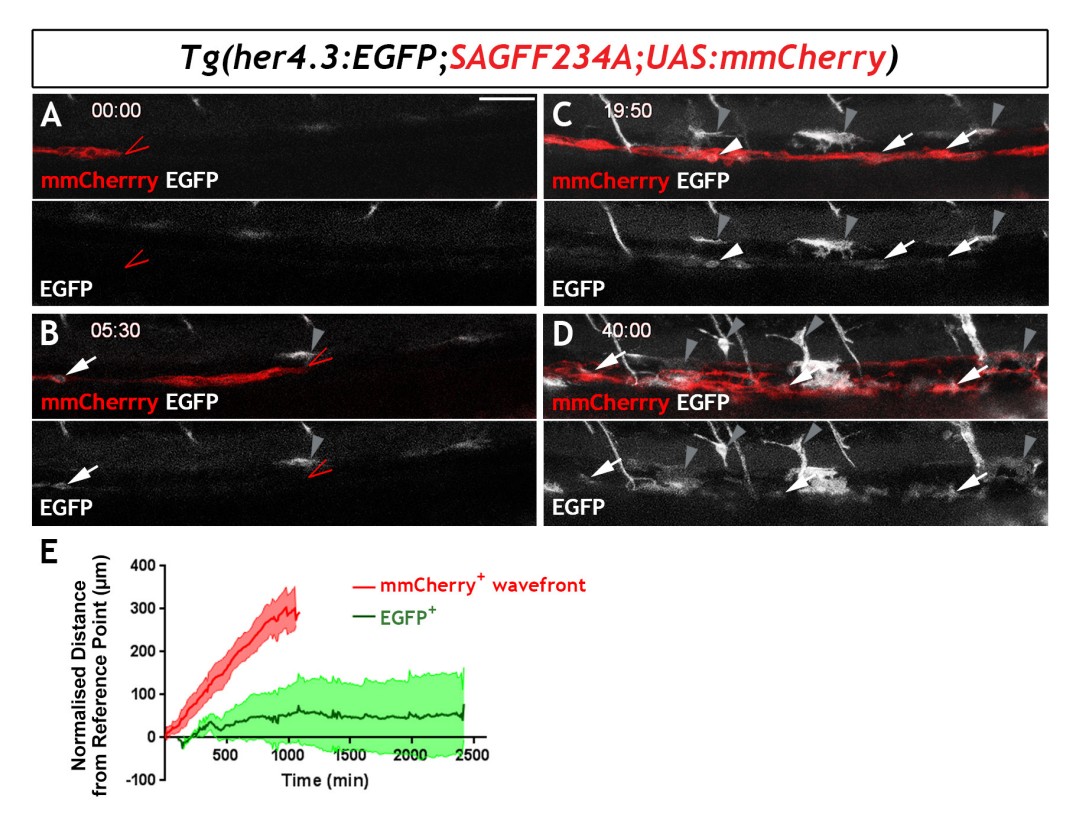

**Figure 5.** Live imaging of *Tg(her4.3:EGFP)*[+] cell ontogenesis in the developing zebrafish ENS. (**A–D**) Still images from time-lapse recording of a *Tg(her4.3:EGFP;SAGFF234A;UAS:mmCherry)* embryo imaged from 56 hpf (denoted as 00:00) until 96 hpf (40:00), a representative example of n = 18 biological replicates. At 00:00 (**A**) the mmCherry[+] wavefront of NC cells (red, red arrowhead) is at the rostral side of the field of view (FOV) and no EGFP[+] cells (grey) are present. At 05:30 (**B**), the first EGFP[+] cells (grey, arrow) appear within the mmCherry[+] NC cell column (red), behind the migratory wavefront. Bright GFP[+] melanocytes are designated (grey arrowheads). (**C**) At 19:50 the NC cell column extends throughout the FOV and the number of EGFP[+] cells (grey, arrows) has increased. White arrowhead points to an EGFP[+] cell exhibiting a rounded morphology, which can be seen to divide in subsequent time lapse images. An increasing number of bright GFP[+] melanocytes appear (grey arrowheads), and are relatively static in the time lapse recordings. (**D**) At the end of the recording (40:00), EGFP[+] cells (grey) can be found throughout the gut (white arrowheads). Abundant brightly GFP[+] melanocytes are present in the gut region (grey arrowheads), whose characteristic morphology is apparent. (**E**) Quantification of cell displacement (normalised distance from reference point/time) of the mmCherry[+] wavefront (red) and EGFP[+] cells (green), data describing 132 cells from four fish. Data are given as mean ± SD. All confocal images are max projections of short confocal stacks. 50 μm scale bar in A.

The online version of this article includes the following figure supplement(s) for figure 5:

**Figure supplement 1.** Lineage analysis reveals that *Tg(her4.3:EGFP)* expressing cells are derived from the embryonic NC cell population that gives rise to the ENS.

Our earlier observation that EGFP expressing cells in the ENS of *Tg(her4.3:EGFP)* zebrafish embryos undergo only a limited number of cell divisions suggested that EdU (5-ethynyl-2'-deoxyuridine) labelling of EGCs in adult animals could be used to trace the progeny of proliferating cells and determine their fate. Consistent with the MCM5 immunostaining, we found that at the end of a 3 day EdU labelling pulse (t0), 8.0 ± 4.3% of GFP[+] cells in the gut of 3 month old *her4.3:EGFP* transgenic zebrafish were co-labelled with EdU (*Figure 6A,B and D*). The similar percentage of MCM5[+] and EdU[+] cells suggests that *Tg(her4.3:EGFP)* cells represent a largely quiescent cell population and the that dividing cells have long cell cycles.

In these experiments, the majority of (GFP[+]EdU[+]) cells formed doublets composed of cells with similar morphology and GFP signal intensity (*Figure 6B* and *Figure 6—figure supplement 2B*). Occasionally, one or both cells in the doublets exhibited reduced GFP signal (*Figure 6—figure supplement 2C,D*), suggesting that, similar to larval stages, the daughters of dividing EGCs in adult *her4.3:EGFP* transgenic zebrafish differentiate into GFP[-] enteric neurons. This idea was supported

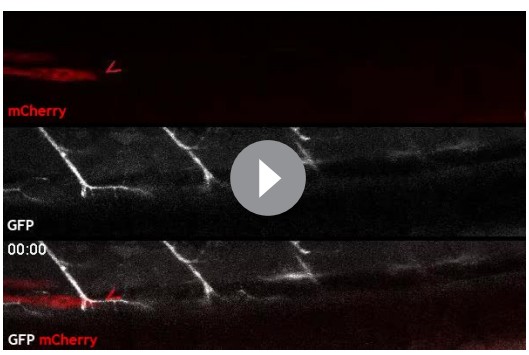

**Video 2.** Representative time-lapse image from a *Tg(her4.3:EGFP;SAGFF234A;UASmmCherry)* embryo. Time-lapse imaging revealed that *Tg(her4.3:EGFP)*⁺ cells (grey, white arrowheads) are found within the mmCherry⁺neural crest cells (red) that are colonising the developing gut, but the EGFP⁺ cells appear behind the wavefront of migration (red arrowheads). Time given is shown as hh:mm from the start of recording. See methods for details.

https://elifesciences.org/articles/56086#video2

by the identification 4 days post-labelling (t4 chase) of EdU⁺ doublets that included GFP⁺-HuC/D⁻ and GFP⁻HuC/D⁺ cells (**Figure 6A,C**). The loss of GFP signal from the daughters of proliferating EGCs cells was also supported by cell population analysis which demonstrated a reduction in the percentage of EdU⁺GFP⁺ cells (t4: $3.6 \pm 3.4\%$, $p=6.01\times10^{-7}$; t11: $3.9 \pm 3.8\%$, $p=7.61\times10^{-6}$) (**Figure 6D**). Interestingly, the reduced percentage of EdU⁺GFP⁺ cells during the EdU chase period was associated with a concomitant increase in the representation of EdU⁺ enteric neurons at t4 ($0.71 \pm 0.80\%$, $p=6.0\times10^{-7}$) and t11 ($0.70 \pm 0.82\%$, $p=1.5\times10^{-6}$) relative to t0 ($0.068 \pm 0.13\%$) (**Figure 6E**). Together, these experiments suggest that the progeny of proliferating EGCs in the zebrafish ENS can differentiate into neurons under physiological conditions.

To provide further evidence in support of the lineage relationship between GFP⁺EdU⁺ cells and newborn enteric neurons (HuC/D⁺EdU⁺), we used confocal microscopy and mathematical modelling to estimate the densities of these cell types within circles of increasing radius centred on EdU⁺ cells (**Figure 6F**; *Tay et al., 2017*). We reasoned that closer proximity of HuC/D⁺EdU⁺ and GFP⁺EdU⁺ cells relative to that expected from random distribution of lineally unrelated cells would indicate origin from common progenitors undergoing cell division. The densities observed at t0, t4 and t11 were compared to values of uniformly distributed cell types generated randomly by Monte Carlo simulations ($>2\times10^{3}$ per sampling time). This analysis revealed that the actual densities of GFP⁺EdU⁺ and HuC/D⁺ EdU⁺ cells were significantly higher within the smaller radius circles (<60 μm from the cell of interest) in comparison to those expected by chance, suggesting that the observed homotypic (GFP⁺EdU⁺/GFP⁺EdU⁺) and heterotypic (GFP⁺EdU⁺/HuC/D⁺EdU⁺) clusters of EdU⁺ ENS cells were descendants of a common proliferating progenitor (**Figure 6G**). EdU⁻ cells exhibited densities similar to those expected in randomly mixed populations (data not shown). This analysis provides further support to the idea that descendants of proliferating *Tg(her4.3:EGFP)* expressing ENS cells are capable of undergoing neuronal differentiation in the gut of adult zebrafish.

Next, we considered the possibility that the GFP⁻ non-neuronal ENS cell population (**Figure 3B**) is also derived from GFP⁺ progenitors and represents an intermediate stage of neurogenic commitment, in a process analogous to the differentiation of GFP⁺ RGCs in the pallium of *her4.3:EGFP* transgenic zebrafish. To examine this, we pulse-labelled 3 month old *Tg(her4.3:EGFP;sox10:Cre;Cherry)* transgenics with EdU (**Figure 6A**) and followed the descendants of proliferating EGCs in the context of the entire ENS lineage. Consistent with our previous analysis (**Figure 6E**), the percentage of enteric neurons labelled by EdU (Cherry⁺HuC/D⁺EdU⁺) at t4 and t11 was higher relative to t0 (t0: $0.021 \pm 0.15\%$; t4: $0.28 \pm 1.2\%$, $p=0.06$; t11: $0.37 \pm 0.95\%$, $p=0.0014$) (**Figure 6H**).

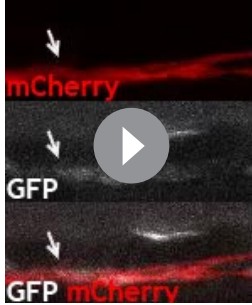

**Video 3.** Representative recording of de novo EGFP expression in time-lapse recording from *Tg(her4.3:EGFP;SAGFF234A;UASmmCherry)* embryos. De-novo *her4.3:EGFP* transgene expression (grey) within the enteric nervous system (red) is observed during time lapse recordings of developing *Tg(her4.1:EGFP;SAGFF234A;UAS:mCherry)* embryos (arrow). https://elifesciences.org/articles/56086#video3

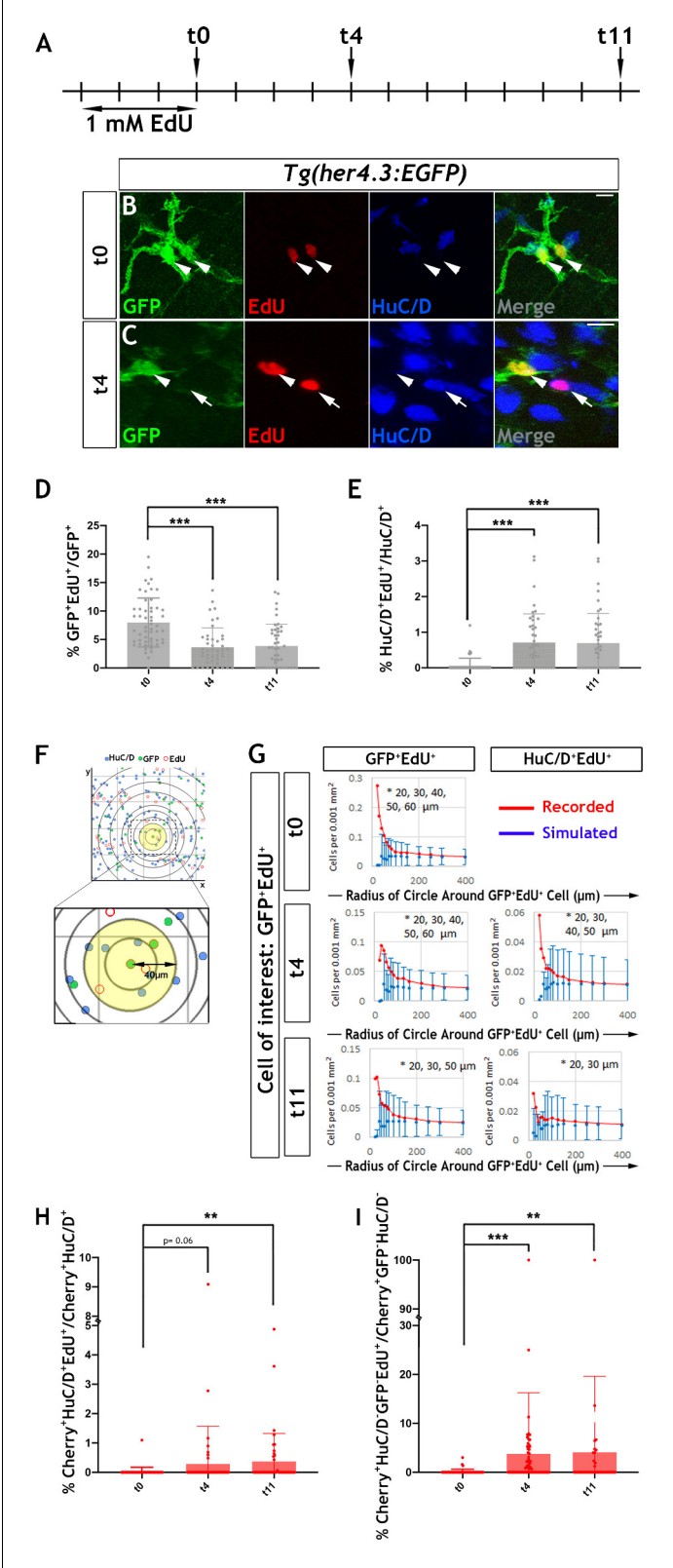

**Figure 6.** Proliferation and neurogenic differentiation of adult her4.3:EGFP[+] ENS cells during homeostasis. (**A**) Schematic representation of experimental design. Adult *Tg(her4.3:EGFP)* zebrafish were immersed in 1 mM EdU for three days and analysed at 0 (t0), 4 (t4) or 11 (t11) days after EdU pulse. (**B–C**) GFP (green) and HuC/D (blue) immunostaining of intestines from EdU (red) pulsed animals harvested at t0 (**B**) and t4 (**C**). Arrowheads (in B and C) point to GFP[+]HuC/D[-]EdU[+] cells. Arrow (in C) indicates a GFP[-]HuC/D[+]EdU[+] neuron. 10 µm scale bars in B-C merge panels. All confocal images

*Figure 6 continued on next page*

*Figure 6 continued*

are max projections of short confocal stacks. (D) Quantification of the percentage of GFP$^+$ cells labelled with EdU at t0, t4 and t11 (mean ± SD). (E) Quantification of the percentage of EdU-labelled enteric neurons at t0, t4 and t11, with biological replicates t0 n = 6, t4 n = 5, t11 n = 5 (mean ± SD.) (F) Strategy for computational analyses of the density of EdU-labelled HuC/D$^+$ and EGFP$^+$ cells. EdU$^+$GFP$^+$ cells were positioned at the centre of concentric circles of increasing radius and the density of EdU$^+$GFP$^+$ and EdU$^+$HuC/D$^+$ cells within each circle was calculated. An example of a 40 μm radius circle (yellow) is shown in higher magnification. (G) Recorded (red graph) and simulated (blue graph) densities of EdU$^+$HuC/D$^+$ and EdU$^+$GFP$^+$ cells (y axis) in concentric circles of increasing radius (x axis) around EdU$^+$GFP$^+$ cells. Monte Carlo simulation of random distribution of EdU$^+$HuC/D$^+$ or EdU$^+$GFP$^+$ cells were performed >2000 times for each dataset in order to establish baseline densities arising in randomly mixed populations. Error bars represent mean ±90% confidence intervals. At all time-points analysed, recorded densities of EdU$^+$HuC/D$^+$ and EdU$^+$GFP$^+$ cells were above the confidence interval (bars) of the simulated densities in 20-60 μm circles (indicated by asterisk). (H, I) Quantification of the percentage of EdU-labelled Cherry$^+$HuC/D$^+$ neurons (H) and Cherry$^+$GFP$^-$HuC/D$^-$ cells (I) at t0, t4 and t11 in the intestine of *her4.3:gfp;sox10:Cre;Cherry* transgenics pulse-labelled with EdU according to the protocol shown in panel A, with biological replicates: t0 n = 6; t4, n = 5; t11 n = 6 (mean ± SD). *p<0.05, **p<0.01, ***p<0.001.

The online version of this article includes the following figure supplement(s) for figure 6:

**Figure supplement 1.** The *Tg(her4.3:EGFP)* cells are actively proliferating in adult homeostasis.
**Figure supplement 2.** Adult *Tg(her4.3:EGFP)* cells take up EdU and appear in doublets.
**Figure supplement 3.** Working model of enteric glia acting as a source of neural progenitors in adult zebrafish during homeostatic conditions.

---

Interestingly, this increase was paralleled by an increased percentage of EdU-labelled GFP$^-$ non-neuronal ENS cells (Cherry$^+$GFP$^-$HuC/D$^-$EdU$^+$) at t4 and t11, relative to t0 (t0: 0.12 ± 0.5%; t4: 3.7 ± 12.5%, p=1.84×10$^{-6}$; t11: 4.1 ± 15.5%, p=0.0024) (*Figure 6I*). Together these studies suggest that loss of *Tg(her4.3:EGFP)* expression in the daughters of proliferating EGCs is likely to reflect neurogenic commitment preceding overt neuronal differentiation.

## Notch signalling regulates the dynamics of EGCs in the gut of adult zebrafish

Inhibition of Notch signalling promotes the proliferation and neurogenic differentiation of *Tg(her4.3:EGFP)* expressing RGCs in the telencephalon of zebrafish (*Alunni et al., 2013*; *Chapouton et al., 2010*). This, together with the observed downregulation of the *her4.3:EGFP* transgene upon neuronal differentiation of GFP$^+$ cells (*Figure 6C*), suggested that canonical Notch activity regulates the proliferation and differentiation dynamics of EGCs in zebrafish. To examine this possibility, we blocked Notch signalling in adult zebrafish by treating them with the γ-secretase inhibitor LY411575 (referred to as LY) (*Alunni et al., 2013*; *Rothenaigner et al., 2011*) for 7 days. To assess the proliferative and neurogenic response of ENS cells, animals were also exposed to EdU during the last 3 days of LY treatment (*Figure 7A*). As expected, LY treatment of *Tg(her4.3:EGFP)* zebrafish resulted in rapid loss of GFP signal from the gut (*Figure 7—figure supplement 1*). Although this experiment confirmed that *Tg(her4.3:EGFP)* is a *bona fide* target of canonical Notch signalling in the ENS, it precluded the use of this transgene as a marker and lineage tracer of the EGC response to LY treatment. Therefore, we applied LY and EdU to *Tg(sox10:Cre;Cherry)* animals and analysed the entire population of non-neuronal ENS cells at the end of the LY/EdU treatment period (t0). As shown in *Figure 7B*, Notch inhibition in 3–4 month old *Tg(sox10:Cre;Cherry)* zebrafish resulted in a dramatic increase in the percentage of non-neuronal ENS cells incorporating EdU (Cherry$^+$HuC/D$^-$EdU$^+$) (control: 0.0387 ± 0.21%; LY: 15.6 ± 17.0%, p=2.67×10$^{-7}$). A robust proliferative response of non-neuronal ENS cells was also observed in 6 month old *Tg(sox10:Cre;Cherry)* animals (control: 0.832 ± 1.87%; LY: 6.95 ± 8.2%, p=1.98×10$^{-5}$) (*Figure 7D*). Interestingly, LY treatment also resulted in increased enteric neurogenesis (Cherry$^+$HuC/D$^+$EdU$^+$ cells) in both 3 month old (control: 0.0330 ± 0.18%; LY: 2.12 ± 7.8%, p=3.70×10$^{-4}$) and 6 month old (control: 0.0652 ± 0.22%; LY: 1.56 ± 3.8%, 3.81 × 10$^{-4}$) animals (*Figure 7C,E*). It remains unclear whether the apparent increase in neurogenesis following LY treatment indicates a direct role of Notch signaling on neuronal differentiation of EGCs or an indirect consequence of their enhanced proliferation. Irrespective of the exact mechanisms, our experiments demonstrate that, similar to pallial RGCs (*Alunni et al., 2013*), Notch signalling regulates the dynamics of EGCs in the vertebrate gut throughout life.

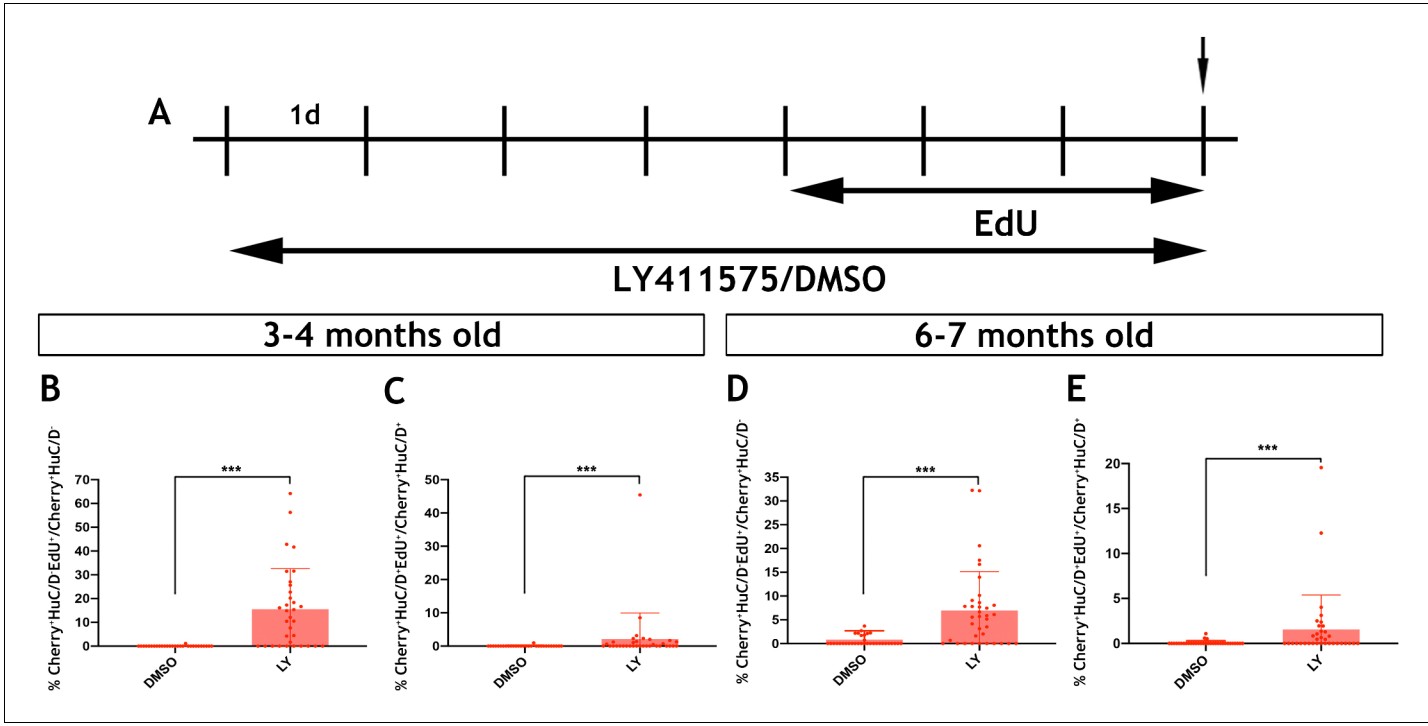

**Figure 7.** Notch signalling regulates the activation and differentiation of adult zebrafish EGCs. (**A**) Schematic representation of experimental protocol for LY/EdU treatment of adult zebrafish. (**B–E**) Quantification of the effect of Notch inhibition on the proliferation (**B** and **D**) and neuronal differentiation (**C** and **E**) of EGCs in 3–4 month old (**B** and **C**) and 6–7 month old (**D** and **E**) animals. N = 4 biological replicates per condition. Data are given as mean ± SD. ***p<0.001.

The online version of this article includes the following figure supplement(s) for figure 7:

**Figure supplement 1.** Notch inhibition in adults leads to loss of GFP expression from the *Tg(her4.3:EGFP)* transgene.

## Discussion

Here, we characterise the non-neuronal compartment of the zebrafish ENS and identify both familiar and unexpected properties of EGCs in teleosts. Specifically, we demonstrate that markers commonly used for the identification of peripheral glial cells in higher vertebrates are not detected in zebrafish EGCs, but that EGCs share morphological features and gene expression programmes with their mammalian counterparts. However, in contrast to mammalian enteric glia, but in accordance with the properties of brain RGCs, the population of zebrafish EGCs is dynamic, undergoing self-renewing proliferation and neuronal differentiation during homeostasis, which are regulated by Notch signalling. Our findings highlight the neural precursor potential of vertebrate enteric glia *in vivo* and reveal previously unanticipated similarities to brain NSCs.

Earlier histological studies demonstrated that mammalian enteric glia are remarkably similar to protoplasmic astrocytes and express the intermediate filament GFAP, a characteristic astrocytic marker (*Jessen and Mirsky, 1980*; *Rühl, 2005*). Further EM analysis revealed diagnostic ultrastructural characteristics of intestinal neuroglia networks in rodents (*Gabella, 1981*). Extending these early reports, we and others have identified four morphological subtypes of mammalian enteric glia, which correlate with their position in the gut and relative to the ganglionic network in the gut wall (*Boesmans et al., 2015*; *Gulbransen and Sharkey, 2012*; *Hanani and Reichenbach, 1994*). Our current experiments demonstrate that all cardinal morphological and ultrastructural features ascribed to mammalian enteric glia are also found in the *Tg(her4.3:EGFP)⁺* non-neuronal compartment of the zebrafish ENS, thus providing strong evidence that it represents the EGC lineage of the teleost ENS. Our failure to detect glial markers commonly used to identify mammalian enteric glia (such as GFAP and S100b) indicates that the expression of these genes may not be integral to the genetic programmes operating in the vertebrate ENS, but rather signifies dynamic physiological states of EGCs adopted in response to specialised local cues. In support of this idea, GFAP is dynamic and is

normally detected in a subpopulation of mammalian EGCs *in vivo* (*Boesmans et al., 2015*) and expression of GFAP and S100b is enhanced in primary cultures of human enteric glia challenged with pro-inflammatory stimuli (*Cirillo et al., 2011*). It would be interesting to determine whether these glial markers are also upregulated in zebrafish EGCs following inflammatory pathology, infection or injury.

Despite the failure to detect canonical glia marker expression, our transcriptomic analysis of zebrafish EGCs revealed a considerable overlap in the gene expression profile of teleost and mammalian enteric glia. Among the genes expressed by both lineages are those encoding the early NC cell markers *sox10*, *foxd3*, *tfap2a*, *zeb2b* and *plp1* (*Hari et al., 2012*; *Knight et al., 2003*; *Mundell and Labosky, 2011*; *Mundell et al., 2012*; *Wang et al., 2011a*; *Weider and Wegner, 2017*), PNS glia-specific marker *col28a1b* (*Grimal et al., 2010*), as well as the stem cell regulators *sox2* and *ptprz1a/b* (*Belkind-Gerson et al., 2017*; *Fujikawa et al., 2017*; *Heanue and Pachnis, 2011*). The roles of these genes have been studied extensively in the context of neural development (*sox10*, *foxd3*, *tfap2a*, *zeb2b*, *vim*, *sox2*) and stem cell dynamics (*sox2*, *ptprz1a/b*), but their potential contribution to the homeostasis and function of enteric glia in adult animals remains unknown. We suggest that the shared gene expression modules we have identified between teleost and mammalian enteric glia represent evolutionary conserved regulatory programmes that are critical for intestinal physiology and ENS homeostasis and highlight the potential of vertebrate EGCs to serve as neurogenic precursors.

One of the unexpected findings of our work is the relatively small size of the non-neuronal compartment in the zebrafish ENS relative to its mammalian counterpart. A series of studies demonstrating that glial cells regulate synaptic activity of CNS neural circuits have led to the suggestion that the enhanced capacity for neural processing of the brain in higher vertebrates has been fuelled during evolution by the increased number, size and complexity of astrocytes (*Han et al., 2013*; *Oberheim et al., 2006*). Perhaps the higher number of enteric glia in mammals, relative to teleosts, may also reflect an increase in the functional complexity of intestinal neural circuits during vertebrate evolution and an enhanced scope of EGCs in the regulation of the complex gut tissue circuitry that maintains epithelial cell homeostasis, host defence and healthy microbiota (*Grubišić and Gulbransen, 2017*).

Several reports have documented that peripheral glial cells can acquire properties of neural crest stem cells (NCSCs) and give rise to diverse cell types. For example, Schwann cell precursors (SCPs) associated with growing nerves in mammalian embryos, in addition to generating the Schwann cell lineage of adult animals, also function as multipotent progenitors giving rise to diverse cell types, including mesenchymal and neuroendocrine cells, parasympathetic neurons and melanocytes (*Parfejevs et al., 2018*; *Petersen and Adameyko, 2017*). Echoing the developmental potential of SCPs, ENS progenitors already expressing molecular markers attributed to EGCs are also capable of generating enteric neurons and mature enteric glia (*Cooper et al., 2016*; *Cooper et al., 2017*; *Lasrado et al., 2017*). In addition to these studies, a growing body of evidence indicates that NCSC properties can be acquired by peripheral glia cell lineages from adult animals, including Schwann cells, glia of the carotid body and EGCs (*Jessen et al., 2015*; *Pardal et al., 2007*). However, it is generally thought that the reprogramming of differentiated glial cells into a NCSC-state is induced by injury, infection or other types of stress, including tissue dissociation and culture. Thus in mammals, EGCs can undergo limited neurogenesis in response to chemical injury to the myenteric plexus, pharmacological activation of serotonin signalling or bacterial gut infection (*Belkind-Gerson et al., 2017*; *Joseph et al., 2011*; *Laranjeira et al., 2011*; *Liu et al., 2009*). By providing evidence that zebrafish EGCs, in addition to their *bona fide* role as glial cells, also serve as constitutive ENS progenitors *in vivo*, our studies argue that the neurogenic potential of mammalian enteric glia disclosed under conditions of injury and stress, reflects an earlier evolutionary state of anamniote vertebrates, in which the same cell type exhibited properties of neural progenitors and mature glia. Although it is currently unclear whether neurogenic potential is a unique property of teleost EGCs, we speculate that peripheral glia lineages in lower vertebrates represent NCSCs that retain their developmental options but adjust to the cellular environment they reside in by acquiring additional specialised functions that contribute to local tissue function and homeostasis. Understanding the transcriptional and epigenetic mechanisms that underpin retention of the NCSC character and simultaneously allow novel functional adaptations during ontogenesis represents an exciting challenge of fundamental biology with practical implications in biomedical research. For example, identification of the

molecular mechanisms that drive neuronal differentiation of enteric glia *in vivo* will facilitate strategies to harness the intrinsic neurogenic potential of mammalian EGCs and restore congenital or acquired deficits of intestinal neural circuits.

By subsuming features of both neural progenitors and glial cells, zebrafish EGCs show remarkable and unexpected parallels to RGCs, NSCs that are distributed widely in teleost brain, reflecting its pronounced neurogenic and regenerative potential (*Alunni and Bally-Cuif, 2016*; *Than-Trong and Bally-Cuif, 2015*), and take on functions normally attributed to astrocytes (*Lyons and Talbot, 2015*). The parallels of RGCs and EGCs are likely to extend beyond a cursory parity imposed by the demands of the resident organs (brain and gut) for continuous growth and specialised glia function, and apply to specific cellular and molecular mechanisms controlling their homeostasis and differentiation. Our proposed model (*Figure 6—figure supplement 3*) depicts key stages of the neurogenic trajectory available to adult zebrafish EGCs, which mirrors the stepwise differentiation of RGCs to pallial neurons (*Than-Trong and Bally-Cuif, 2015*; *Than-Trong et al., 2018*). Thus, similar to RGCs, the majority of EGCs remain quiescent at steady state (qEGCs), but in response to as yet unknown signals, a proportion of them enters the cell cycle giving rise to active EGCs (aEGCs). Whether the ability to enter the cell cycle is a property restricted to a subpopulation of qEGCs is currently unknown. More generally, to what extent qEGCs can be subdivided molecularly into subsets with distinct properties and function is an interesting question for future studies. Both qEGCs and aEGCs are currently identified by the Notch activity reporter *Tg(her4.3:EGFP)* and represent reversible cellular states distinguished by cell cycle marker expression and thymidine analogue incorporation (aEGCs). Extinction of *Tg(her4.3:EGFP)* expression is associated with irreversible commitment of aEGCs to enteric neural progenitors (eNPs), which eventually differentiate to mature enteric neurons capable of integration into functional intestinal neural circuits. We suggest that this transient population of eNPs correspond to the HuC/D⁻GFP⁻ cells identified in the ENS of *Tg(her4.3:EGFP;sox10:Cre;Cherry)* zebrafish (*Figure 3B*). The proposed scheme ensures the long-term maintenance of the original population of EGCs and the generation of new enteric neurons to cater for the physical growth of the gut and the plasticity of its intrinsic neural networks.

Previous studies have established the central role of Notch signalling and its target genes in controlling the dynamics of NSCs in vertebrates (*Chapouton et al., 2010*; *Imayoshi et al., 2010*) and uncovered the differential contributions of distinct Notch receptors in regulating RGC proliferation and differentiation in the germinal zones of the zebrafish brain (*Alunni et al., 2013*; *Than-Trong et al., 2018*). Although the relevant Notch signalling components remain to be identified, our experiments provide evidence that the activation and differentiation of EGCs in adult zebrafish gut is also under the control of Notch signalling, pointing to further fundamental similarities in the mechanisms controlling the homeostasis of the CNS and ENS in vertebrates. Notch signalling has also been implicated in the development of the mammalian ENS by inhibiting the intrinsic neurogenic programme of ENS progenitors (*Okamura and Saga, 2008*). Our demonstration that the Notch activity reporter *Tg(her4.3:EGFP)* is activated in ENS progenitors shortly after they invade the gut and initiate neurogenic differentiation suggests a similar role of Notch signalling in the development of the zebrafish ENS, namely attenuation of the strong neurogenic bias of early ENS progenitors acquired as they enter the foregut and induce strong neurogenic transcription factors, such as Phox2B and Ascl1 (*Charrier and Pilon, 2017*).

The detailed hierarchical relationships of the identified cell types in the non-neuronal compartment of the zebrafish ENS and the potential regional differences in the dynamics of EGCs in zebrafish gut remain to be characterised. In addition, to what extent EGC-driven adult enteric neurogenesis in zebrafish depends on regulatory genes that control the differentiation of enteric neurons during vertebrate development (such as *ret*, *ascl1*, *phox2b*) is currently unclear. Nevertheless, the systematic characterization of the molecular programs underpinning the neuronal differentiation of EGCs in adult zebrafish is likely to inform strategies for the activation of the intrinsic neurogenic potential of mammalian EGCs and the repair of gastrointestinal neural networks damaged by disease or aging.

# Materials and methods

## Key resources table

| Reagent type (species) or resource | Designation | Source or reference | Identifiers | Additional information |
|---|---|---|---|---|
| Genetic reagent (*Danio rerio*) | *Tg(SAGFF234A)* | *Asakawa et al., 2008*; *Kawakami et al., 2010* | SAGFF (LF)234A | zTrap Resource from Koichi Kawakami Lab |
| Genetic reagent (*Danio rerio*) | *Tg(UAS:GFP)* | *Kawakami et al., 2010* | | Resource from Koichi Kawakami Lab |
| Genetic reagent (*Danio rerio*) | *Tg(−4.7sox10:Cre)* | *Rodrigues et al., 2012* | Tg(−4.7sox10:Cre)ba74 | |
| Genetic reagent (*Danio rerio*) | *Tg(βactin-LoxP-STOP-LoxP-hmgb1-mCherry)* | *Wang et al., 2011b* | Tg(bactin2:loxP-STOP-loxP-hmgb1-mCherry)jh15 | |
| Genetic reagent (*Danio rerio*) | *ret^hu2846* | ZIRC; *Knight et al., 2011* | ZL3218 | |
| Genetic reagent (*Danio rerio*) | *Tg(gfap:GFP)* | ZIRC; *Bernardos and Raymond, 2006* | ZL1070 | |
| Genetic reagent (*Danio rerio*) | *Tg (−3.9nestin:GFP)* | EZRC; *Lam et al., 2009* | 15206 | |
| Genetic reagent (*Danio rerio*) | *Tg(her4.3:EGFP)* | *Yeo et al., 2007* | | ZDB-ALT-070612–3 |
| Genetic reagent (*Danio rerio*) | *Tg(SAGFF217B)* | *Kawakami et al., 2010* | | zTrap Resource from Koichi Kawakami Lab |
| Genetic reagent (*Danio rerio*) | *Tg(UAS:mmCherry)* | this paper | | |
| Antibody | anti-HuC/D (Mouse monoclonal) | Thermofisher | A21272; RRID:AB_2535822 | 1:200 |
| Antibody | anti-Cherry (Goat polyclonal) | Antibodies online | ABIN1440057 | 1:500 |
| Antibody | anti-GFP (Chick polycloonal) | Abcam | ab13970; RRID:AB_300798 | 1:500 |
| Antibody | anti-S100ß (Rabbit polyclonal) | Dako | Z0311; RRID:AB_10013383 | 1:500 |
| Antibody | anti-mouse GFAP (Rabbit polyclonal) | Sigma | G9269; RRID:AB_477035 | 1:500 |
| Antibody | anti-zebrafish GFAP (Rabbit polyclonal) | Genetex | GTX128741; RRID:AB_2814877 | 1:500 |
| Antibody | zrf-1 anti-zebrafish GFAP (Mouse mononclonal) | Abcam | ab154474; RRID:AB_10013806 | 1:200 |
| Antibody | anti-BFABP (Rabbit polyclonal) | Merck | ABN14; RRID:AB_10000325 | 1:500 |
| Antibody | anti-AcTu (Mouse monoclonal) | Sigma | T6793; RRID:AB_477585 | 1:1000 |
| Antibody | anti-MCM5 | gift from Soojin Ryu | | 1:500 |
| Commercial assay or kit | RNAscope Flourescent Multiplex Kit | Advanced Cell Diagnostics | 320850 | |

*Continued on next page*

*Continued*

| Reagent type (species) or resource | Designation | Source or reference | Identifiers | Additional information |
|---|---|---|---|---|
| Commercial assay or kit | RNAscope Probe-Dr-sox10 | Advanced Cell Diagnostics | 444691-C3 | |
| Commercial assay or kit | RNAscope Probe-Dr-foxd3 | Advanced Cell Diagnostics | 444681-C3 | |
| Commercial assay or kit | RNAscope Probe-Dr-ret | Advanced Cell Diagnostics | 579531 | |
| Chemical compound, drug | Notch inhibitor LY411575 | Cambridge Bioscience | 16162 | |

## Animals

All animal experiments were carried out in compliance with the Animals (Scientific Procedures) Act 1986 (UK) and in accordance with the regulatory standards of the UK Home Office (Project Licence PCBBB9ABB). Experimental protocols were approved by the local Animal Welfare and Ethical Review Body (AWERB) of the Francis Crick Institute. Zebrafish stocks were maintained as described (*Heanue et al., 2016a*; *Westerfield, 2000*). Embryos and larvae were maintained and staged as described (*Heanue et al., 2016a*), while embryos used for time lapse were reared in 0.2 mM PTU from 24 hpf to inhibit melanisation, as described (*Westerfield, 2000*). Transgenic and mutant lines used were as follows: *Tg(SAGFF234A)* (*Asakawa et al., 2008*; *Kawakami et al., 2010*); *Tg(UAS:GFP)* (*Kawakami et al., 2010*), *Tg(−4.7sox10:Cre)* (*Rodrigues et al., 2012*), *Tg(βactin-LoxP-STOP-LoxP-hmgb1-mCherry)* (*Wang et al., 2011b*), *ret*[hu2846] (*Knight et al., 2011*), *Tg(gfap:GFP)* (*Bernardos and Raymond, 2006*), *Tg (−3.9nestin: GFP)* (*Lam et al., 2009*), *Tg(her4.3:EGFP)* (*Yeo et al., 2007*), *Tg(SAGFF217B)* (*Kawakami et al., 2010*). Note that the *Tg(her4.3:EGFP)* designation is the current ZFIN reference for this transgene, however it is also variously referred to as *Tg(her4:EGFP)* (*Yeo et al., 2007*) or *Tg (her4.1GFP)* (*Kizil et al., 2012*). *her4.3* is one of 6 (of 9) mammalian orthologues of mammalian *Hes5* found in tandem duplication on chromosome 23 of the zebrafish genome (*Zhou et al., 2012*). The stable *Tg(UAS:mmCherry)* line was generated by Tol2 transgenesis: co-microinjection of TOL2 transposase with a construct containing membrane-mCherry (mmCherry) downstream of two copies of the Gal4 recognition sequence UAS, with bicistronic α crystalinP:RFP cassette enabling red eye selection of carriers, as described previously (*Gerety et al., 2013*). Genotyping was done based on the lines' previously described distinct fluorescent patterns, or by PCR in the case of *Tg(ret* [hu2846/+]*)*, as described (*Knight et al., 2011*).

## Immunohistochemistry

Immunohistochemistry was performed as previously described on whole-mount embryos/larvae or whole-mount adult intestines and brains (*Heanue et al., 2016b*). Primary antibodies used were as follows: HuC/D (mouse, ThermoFisher A21272, 1:200), Cherry (goat, Antibodies online ABIN1440057, 1:500), GFP (chick, Abcam ab13970, 1:500), S100β (rabbit, Dako Z0311, 1:500), mGFAP (rabbit, Sigma G9269, 1:500), zGFAP (rabbit, Genetex GTX128741, 1:500), zrf-1 (mouse, Abcam ab154474, 1:200), BFABP (Merck ABN14, 1:500), AcTu (mouse, Sigma T6793, 1:1000), MCM5 antibody (1:500, kindly provided by Soojin Ryu, Max Planck Institute for Medical Research, Heidelberg, Germany) and appropriate secondary antibodies conjugated to AlexaFluor 405, 488, 568 and 647 were used for visualisation (Molecular Probes). EdU was developed using the EdU Click-it kit following the manufacturer's instructions and combined with fluorophores Alexa555 or Alexa647 (C10337 and C10339). For MCM5 labelling, antigen retrieval was required to expose the epitope. Briefly, after immunostaining for GFP, antigen retrieval with Citrate buffer (pH 6.0) was performed. All tissues were mounted on Superfrost Plus slides with Vectashield Mounting Media with/ without DAPI (H1200/H1000, respectively). In all experiments, the CNS regions (larval brain and spinal cord or adult brain) provide a positive control (i.e *Figure 1—figure supplement 1J–O*) and negative controls are provided by immunostaining without primary antibody. Immunohistochemistry

images were captured on a Leica CM6000 confocal microscope or an Olympus FV3000 confocal microscope, with standard excitation and emission filters for visualising DAPI, Alexa Flour 405, Alexa Flour 488, Alexa Flour 568 and Alexa Flour 647. Orthogonal views are used to clearly identify cells as expressing a marker of interest. Images processed with Adobe Photoshop 8.

## Purification of ENS nuclei from adult gut muscularis externa

Adult *Tg(sox10:Cre;Cherry)* zebrafish intestines were first dissected, then cut along their length and immersed in HBSS (no calcium, no magnesium, (ThermoFisher 14170088)) containing 20 mM EDTA and 1% Penicillin/Streptomycin (ThermoFisher, 15140122) for 20–25 min at 37 ˚C until the epithelia cell layer was seen to begin detaching from the overlying muscularis externa, evident by clouding of the HBSS solution. After several washes in PBS (ThermoFisher 14190094), the tissue was placed under a dissecting microscope and the muscularis externa was grasped in forceps and agitated briefly to detach any remaining associated epithelial cells. Muscularis externa was tranfered to a fresh tube and purification of nuclei was performed essentially as described (*Obata et al., 2020*). Briefly, dounce homogenization was performed in lysis buffer (250 mM sucrose, 25 mM KCl, 5 mM MgCl$_2$, 10 mM Tris buffer with pH8.0, 0.1 mM DTT) containing 0.1% Triton-X, cOmplete EDTA-free protease inhibitor (Sigma-Aldrich) and DAPI. The homogenate was filtered to remove debris and centrifuged to obtain a pellet containing the muscularis externa nuclei. For flow cytometric analysis, doublet discrimination gating was applied to exclude aggregated nuclei, and intact nuclei were determined by subsequent gating on the area and height of DAPI intensity. Both mCherry$^+$ and mCherry$^-$ nuclear populations (termed Cherry$^+$ and Cherry$^-$ in text and figures) were collected directly into 1.5 mL tube containing Trizol LS reagent (Invitrogen) using the Aria Fusion cell sorter (BD Biosciences). The obtained FCS data were further analysed using FlowJo software version 10.6.1. For each replicate, sorted cells from an average of 30 adult guts were pooled, containing approximately 30,000 mCherry$^+$ or mCherry$^-$ nuclear populations.

## RNA-sequencing and bioinformatic analysis

RNA was isolated from nuclei populations using the PureLink RNA Micro Kit (Invitrogen #12183016), according to the manufacturer's instructions. Double stranded full-length cDNA was generated using the Ovation RNA-Seq System V2 (NuGen Technologies, Inc). cDNA was quantified on a Qubit 3.0 fluorometer (Thermo Fisher Scientific, Inc), and then fragmented to 200 bp by acoustic shearing using Covaris E220 instrument (Covaris, Inc) at standard settings. The fragmented cDNA was then normalized to 100 ng, which was used for sequencing library preparation using the Ovation Ultralow System V2 1–96 protocol (NuGen Technologies, Inc). A total of 8 PCR cycles were used for library amplification. The quality and quantity of the final libraries were assessed with TapeStation D1000 Assay (Agilent Technologies, Inc). The libraries were then normalized to 4 nM, pooled and loaded onto a HiSeq4000 (Illumina, Inc) to generate 100 bp paired-end reads.

## Bioinformatics method summary RNA-sequencing-analysis

Sequencing was performed on an Illumina HiSeq 4000 machine. The 'Trim Galore!' utility version 0.4.2 was used to remove sequencing adaptors and to quality trim individual reads with the q-parameter set to 20 (https://www.bioinformatics.babraham.ac.uk/projects/trim_galore/). Then sequencing reads were aligned to the zebrafish genome and transcriptome (Ensembl GRCz10 release-89) using RSEM version 1.3.0 (*Li and Dewey, 2011*) in conjunction with the STAR aligner version 2.5.2 (*Dobin et al., 2013*). Sequencing quality of individual samples was assessed using FASTQC version 0.11.5 (https://www.bioinformatics.babraham.ac.uk/projects/fastqc/) and RNA-SeQC version 1.1.8 (*DeLuca et al., 2012*). Differential gene expression was determined using the R-bioconductor package DESeq2 version 1.14.1 (*Love et al., 2014*; *R Development Core Team, 2008*). R: A language and environment for statistical computing. R Foundation for Statistical Computing, Vienna, Austria. ISBN 3-900051-07-0, URL (http://www.R-project.org). Data deposited at NCBI Geo (GSE145885). For differential gene expression analyses, the Wald-test and log-fold shrinkage was used in the context of the DESeq2 R-package (Parameters 'test' of the DESeq2-function was set to 'Wald' and the parameter 'betaPrior' was set to 'TRUE') (*Love et al., 2014*). Gene set enrichment analysis (GSEA) was conducted as described in *Subramanian et al., 2005*. For

conversion from mouse to zebrafish gene names we used the Ensembl biomart tool (http://www.ensembl.org/biomart/martview).

## RNA-Seq literature data gene list integration

For comparison of our transcriptomic data to published zebrafish data (*Roy-Carson et al., 2017*), we utilised the gene list presented in this paper ('upregulated genes' from *Supplementary file 1*) which represents the zebrafish larval ENS neuron transcriptome. For comparison of our data to genes previously described as characterising mammalian ENS glia (*Rao et al., 2015*), we used the list of genes identified in *Rao et al., 2015* Table 1 ('Table 1 Top 25 genes enriched in PLP1[+] enteric glia'), and manually curated the zebrafish orthologues (see *Supplementary file 7*). To identify mouse ENS neuron and ENS glia signature genes we obtained mouse single-cell data from *Zeisel et al., 2018*. Specifically, we downloaded the single-cell read count data file for the enteric cells in the above project from https://storage.googleapis.com/linnarsson-lab-loom/l1_enteric.loom on the 12th Nov 2019. The data were processed using the Seurat package using the standard workflow (*Stuart et al., 2019*). The resolution parameter in the FindClusters function was set to 0.3. Neuronal and glia clusters were identified on the basis of the signature genes Elavl3, Elavl4, Prph (neuron) and Sox10, S100b, and Gfap (glia) (See *Figure 2—figure supplement 3*). The summarized neuron and glia clusters were subjected to a differential gene expression analysis using the FindMarkers Seurat function using the MAST algorithm.

## Fluorescence *in situ* hybridization

Adult zebrafish intestines from *Tg(sox10:Cre;Cherry)* or *Tg(her4.3:EGFP)* were first dissected, then cut along their length, pinned to a silguard plate and immersed in HBSS (ThermoFisher 14170088) containing 20 mM EDTA and 1% penicillin/streptomycin (ThermoFisher, 15140122) for 20–25 min at room temperature to detach the epithelia layer. After several washes in PBS (ThermoFisher 14190094), the epithelia was manually teased away with forceps. After washing in PBS, 4% PFA was added to the plate with pinned tissue to fix overnight at 4 ˚C. Fluorescence *in situ* hybridization was then performed using the Advanced Cell Diagnostics RNAscope Fluorescent Multiplex Kit (ACD #320850), according to manufacturer's specification and essentially as described (*Obata et al., 2020*). Briefly, tissue was washed in PBS, dehydrated through an ethanol series and then incubated with RNAscope Protease III for 25 min. Tissue was incubated overnight at 40˚C in a HybeOven with customized probes (*sox10*, *foxd3*, *ret*). The next day, the tissue was washed twice with Wash Buffer before hybridization the with pre-amplifier, the appropriate amplifier DNA (Amp 1-FL, Amp 2-FL and Amp 3-FL) and appropriate fluorophores (Amp4 Alt A-FL/AltC-FL) at 40˚C for 15–30 min, as per the manufacturer's instructions. Tissues were then processed for immunohistochemistry and mounted directly onto Superfrost Plus slides (ThermoFisher Scientific #10149870) Vectashield Mounting Media without DAPI (VectorLabs H1000). Image were captured on a Leica CM6000 confocal microscope or an Olympus FV3000 confocal microscope, with standard excitation and emission filters for visualising DAPI, Alexa Flour 405, Alexa Flour 488, Alexa Flour 568 and Alexa Flour 647 and images processed with Adobe Photoshop 8.

## Correlative light and electron microscopy

Intestines were dissected from *Tg(her4.3:EGFP;SAGFF217B;UAS:mmCherry)* adult animals and fixed in 4% formaldehyde 0.1% glutaraldehyde in phosphate buffer (PB) overnight at 4˚C. Subsequently, the intestines were sectioned to 150 µm on a Leica vibratome, and stored in 2% formaldehyde in PB. Mid-gut sections were mounted in PB on SuperFrost Plus slides and imaged with an inverted Zeiss 880 confocal microscope with AiryScan, using standard emission and excitation filters for EGFP and mmCherry. A low magnification overview image was acquired using a 20x objective before 2–3 regions of interest (ROI) were identified per section that contained at least one EGFP[+] cell of interest. The Airyscan was aligned for EGFP and mmCherry using an area outside of the ROIs where both fluorophores were identified. After Airyscan alignment, the ROIs were captured using a x63 glycerol objective and pixel size, z-depth and zoom (>1.8 x) were defined by Nyquist's theorem. For super-resolution images, two adult midguts were scanned at low magnification to identify six regions regions of interest (ROIs) containing EGFP[+] or Cherry[+] cells, and six super resolution images taken. Once fluorescence microscopy was completed, the vibratome slices were further fixed in 2.5%

glutaraldehyde and 4% formaldehyde in 0.1 M phosphate buffer (pH 7.4), and processed according to the method of the National Centre for Microscopy and Imaging Research (*Deerinck et al., 2010*) NCMIR methods for 3D EM: a new protocol for preparation of biological specimens for serial block face scanning electron microscopy (https://ncmir.ucsd.edu/sbem-protocol) before flat embedding between sheets of Aclar plastic. CLEM analysis included herein is taken from a single gut slice, from one of two original ROIs for this slice. This ROI contained six GFP$^+$ cell bodies and two Cherry$^+$ cell bodies.

## SBF SEM data collection and image processing

Serial blockface scanning electron microscopy (SBF SEM) data were collected using a 3View2XP (Gatan, Pleasanton, CA) attached to a Sigma VP SEM (Zeiss, Cambridge). Flat embed vibratome slices were cut out and mounted on pins using conductive epoxy resin (Circuitworks CW2400). Each slice was trimmed using a glass knife to the smallest dimension in X and Y, and the surface polished to reveal the tissue before coating with a 2 nm layer of platinum. Backscattered electron images were acquired using the 3VBSED detector at 8,192*8,192 pixels with a dwell time of 6 μs (10 nm reported pixel size, horizontal frame width of 81.685 μm) and 50 nm slice thickness. The SEM was operated at a chamber pressure of 5 pascals, with high current mode inactive. The 30 μm aperture was used, with an accelerating voltage of 2.5 kV. A total of 1,296 images were collected, representing a depth of 64.8 μm, and volume of 432,374 μm$^3$. Downstream image processing was carried out using Fiji (*Schindelin et al., 2012*). The images were first batch converted to 8-bit tiff format, then denoised using Gaussian blur (0.75 pixel radius), and resharpened using two passes of unsharp mask (10 pixel radius 0.2 strength, 2 pixel radius 0.4 strength), tailored to suit the resolution and image characteristics of the dataset. Image registration was carried out using the 'align virtual stack slices' plugin, with a translation model used for feature extraction and registration. The aligned image stacks were calibrated for pixel dimensions, and cropped to individual regions of interest as required. To generate a composite of the two volumes, Bigwarp (*Bogovic JA et al., 2015*; *Russell et al., 2017*) was used to map the fluorescence microscopy volume into the electron microscopy volume which was reduced in resolution to isotropic 50 nm voxels to reduce computational load. The multi-layered cellular composition of the tissue was noted to have caused substantial nonlinear deformation during processing of the sample for electron microscopy when compared to prior fluorescence microscopy. After exporting the transformed light microscopy volume from Bigwarp, a two pixel Gaussian blur was applied, the datasets were combined, and the brightness/contrast adjusted for on-screen presentation. False coloured images were composed by annotating separate semi-transparent layers in Adobe Photoshop CC 2015.5 with reference to prior fluorescence microscopy and 3-dimensional context within the image stack. Only processes that could be clearly tracked through the volume from definitively marked cell bodies were coloured. Cell soma dimensions were determined by finding the largest X and Y extent of the cell body when scrolling through the CLEM volume, determining the entire extent of the cell body in Z, and calculating volume assuming an ellipsoid shape. Cell processes were tracked as far as possible through the CLEM volume. If a direct connection to the cell body was still visible at this farthest distance, the process length was traced on the image back to the middle of the cell soma. If no connection was apparent as was particularly the case for longer distance extensions, the distance in a straight line to the cell body was calculated in XYZ.

## Time lapse imaging of zebrafish larvae

Embryos were raised in 0.2 mM PTU, lightly anaesthetised with 0.15 mg/ml Tricaine, and mounted into embryo arrays and overlaid with 0.6% low melt temperature agarose in embryo media essentially as described (*Heanue et al., 2016a*; *Megason, 2009*). Once set, the mould was overlaid with embryo media containing 0.15 mg/ml Tricane and 0.15 M PTU, and was replaced at least every 24 hr. Larvae were imaged using a Leica CM6000 confocal microscope, with a 20X water dipping objective. Standard excitation and emission filters were used to visualise EGFP and mmCherry expression. For each individual embryo, 33 z-stacks (z thickness 2.014 μm) were collected at a frame rate of 602 s, for 40.333 hr. Cells from the time-lapse recordings were tracked manually using the MTrack2 plugin on Fiji. To correct for growth or movement during the imaging process a reference point was

taken, for each animal, as the point the anterior most spinal nerve, visible in the field of view, touched the gut. All calculated distances were given relative to this reference point.

### EdU labelling

To label proliferative cells, adult zebrafish were kept in system water with 1 mM of EdU (0.05% DMSO) for 72 hr at a density of 4 zebrafish/litre. During chase periods adult zebrafish were kept in system water, which was changed every 2–3 days.

### Mathematical modelling

Since the zebrafish ENS is largely confined to the myenteric plexus, and hence the zebrafish ENS resides within a two dimensional plane, therefore, only X and Y coordinates were used for subsequent analysis. Each image covered a 450 µm-450 µm area and XY coordinates of individual cells were taken as the centre of the nucleus and obtained from the CellCounter plugin for Fiji. We first estimated the density of specifically labelled cells at several distances around every cell type of interest using confocal images with an area 450 µm x 450 µm. Cell density was estimated in circles of increasing radius, $r \in (20, 30, 40,. ., 100, 150,. .,500 \mu m)$, by dividing the number of cells within the circle by the surface area of the circle included within the image. When the radius was larger than the distance of the cell to the image edge, the area of the circle section overlapping with the image was numerically estimated by Monte Carlo simulation methods. We performed 50 Monte Carlo simulations for each confocal image with the observed number of cells of each phenotype in rearranged locations, according to a uniform distribution, on the 450 µm x 450 µm square area. Cell densities were estimated for each simulation as described above. To compare the recorded and simulated densities, we estimated the 90% confidence interval for simulated cell density under the assumption of cell homogeneity by fitting the gamma distribution function to the simulated values. When the average of the measured cell densities lied outside the 90% confidence interval, the observed spatial location was considered to be a non-chance event in a homogenous mixture of cells.

### Notch inhibition

Notch signalling was inhibited by immersion with 10 µM LY411575 (Cambridge bioscience, 16162) (0.04% DMSO) in the system water, and was changed every 2–3 days, control zebrafish were incubated with the equivalent concentration of DMSO (0.04%).

### Counting and statistics

In all experiments, the number of biological replicates (n: individuals, processed independently) is indicated in the figures. For quantifications of cells in embryos and larvae, the entire gut length was quantified. For quantification of cells in adults, nine random regions were counted: three from each of the main gut regions (intestinal bulb, midgut, hindgut). Orthogonal views are used to clearly identify cells as expressing a marker of interest. Cell counting analysis was carried out using the Cell Counter plugin. Statistical analysis was performed using R 3.3.1. Due to the non-normality of most of the data, all comparisons were carried out using a two-sided Mann-Whitney non-parametric test. Resultant p-values were corrected for multiple testing using the Benjamini-Hochberg method as implemented by the p.adjust function. A Pvalue of $\leq 0.05$ was deemed to be significant and in figures designation of graded significance was as follows: $p > 0.05$ (ns = non significant), $p \leq 0.05$ (*), $p \leq 0.01$ (**), $p \leq 0.01$ (***).

## Acknowledgements

We are grateful to Laure Bally-Cuif, Alessandro Alluni, Emmanuel Than-Trong for providing *Tg(her4.3:EGFP)* transgenic fish and specialist knowledge, to Donald Bell for expert advice in time-lapse imaging experiments, to the Aquatics BRF staff for fish husbandry, to the Flow Cytometry STP for FACS cell sorting, to the Advanced Sequencing Facility STP for library prep and sequencing, and to Joe Brock for scientific illustrations.

## Additional information

### Competing interests

Carmen Pin: Carmen Pin is affiliated with AstraZeneca. The author has no financial interests to declare. The other authors declare that no competing interests exist.

### Funding

| Funder | Grant reference number | Author |
|---|---|---|
| Francis Crick Institute | core funding | Vassilis Pachnis |
| BBSRC | BB/L022974/1 | Vassilis Pachnis |

The funders had no role in study design, data collection and interpretation, or the decision to submit the work for publication.

### Author contributions

Sarah McCallum, Yuuki Obata, Conceptualization, Formal analysis, Validation, Investigation, Visualization, Methodology, Writing - original draft, Writing - review and editing; Evangelia Fourli, Formal analysis, Validation, Investigation, Visualization, Writing - original draft, Writing - review and editing; Stefan Boeing, Resources, Data curation, Software, Formal analysis, Validation, Visualization, Methodology, Writing - original draft, Writing - review and editing; Christopher J Peddie, Formal analysis, Validation, Investigation, Visualization, Methodology, Writing - original draft, Writing - review and editing; Qiling Xu, Robert N Kelsh, Resources, Methodology, Writing - review and editing; Stuart Horswell, Data curation, Software, Formal analysis, Visualization, Writing - review and editing; Lucy Collinson, Formal analysis, Supervision, Methodology, Project administration, Writing - review and editing; David Wilkinson, Resources, Supervision, Methodology, Writing - review and editing; Carmen Pin, Resources, Software, Formal analysis, Validation, Investigation, Visualization, Methodology, Writing - original draft, Writing - review and editing; Vassilis Pachnis, Conceptualization, Supervision, Funding acquisition, Writing - original draft, Project administration, Writing - review and editing; Tiffany A Heanue, Conceptualization, Formal analysis, Supervision, Validation, Investigation, Visualization, Methodology, Writing - original draft, Project administration, Writing - review and editing

### Author ORCIDs

Sarah McCallum https://orcid.org/0000-0003-2156-3219
Yuuki Obata https://orcid.org/0000-0001-5461-3521
Evangelia Fourli https://orcid.org/0000-0002-6549-342X
Stefan Boeing https://orcid.org/0000-0003-0495-5659
Stuart Horswell https://orcid.org/0000-0003-2787-1933
Robert N Kelsh https://orcid.org/0000-0002-9381-0066
Lucy Collinson https://orcid.org/0000-0003-0260-613X
David Wilkinson http://orcid.org/0000-0001-6757-7080
Carmen Pin https://orcid.org/0000-0001-8734-6167
Vassilis Pachnis https://orcid.org/0000-0001-9733-7686
Tiffany A Heanue https://orcid.org/0000-0002-6678-8246

### Ethics

Animal experimentation: All animal experiments were carried out in compliance with the Animals (Scientific Procedures) Act 1986 (UK) and in accordance with the regulatory standards of the UK Home Office (Project Licence PCBBB9ABB). Experimental protocols were approved by the local Animal Welfare and Ethical Review Body (AWERB) of the Francis Crick Institute.

### Decision letter and Author response

Decision letter https://doi.org/10.7554/eLife.56086.sa1
Author response https://doi.org/10.7554/eLife.56086.sa2

# Additional files

## Supplementary files

• Supplementary file 1. Table containing the analysed data from the adult zebrafish gut transcriptome comparing expression in Cherry$^+$ vs Cherry$^-$ populations. Log fold change (logFC) of Cherry$^+$ vs Cherry$^-$ populations is shown in column F (logFC_PE_SOX10_vs_PE_neg) and adjusted p-value (padj) is shown in column H. Significant differentially expressed genes were taken as those with logFC >2 or < −2, and padj ≤0.5. Gene names and Ensembl gene IDs found in columns A and B, respectively. See graphical depiction of this data in the volcano plot in *Figure 2A*.

• Supplementary file 2. Table containing the order of heatmap genes and values for *Figure 2—figure supplement 1F*. Genes displayed in the heat map depicting the nRNASeq data of this study were selected as follows: genes with a logFC (Cherry$^+$ vs Cherry$^-$)>0, padj (Cherry$^+$ vs Cherry$^-$)<0.05 and an average TPM of 3. We intersected this selection with the 2,561 genes identified in 'Additional File 2: *Supplementary file 1* of *Roy-Carson et al., 2017* as upregulated in 7 dpf phox2b:EGFP$^+$ gut cells relative to EGFP$^-$ gut. This selection highlights 758 genes depicted in *Figure 2—figure supplement 1F*. Gene names and Ensembl gene IDs found in column K.

• Supplementary file 3. Comparison of the Cherry$^+$ transcriptomic dataset to a single cell transcriptomic dataset of mouse ENS neurons and glia. Comparison of the Cherry$^+$ transcriptomic dataset to a single cell transcriptomic dataset published by *Zeisel et al., 2018*, describing mouse ENS neuronal and glial transcriptomes. Genes differentially expressed in the Zeisel dataset were determined as described in the Materials and Methods (logFC >0.2 in neurons vs. glia or glia vs. neurons, and p-value≤0.05). Orthologues of those genes were determined using Ensembl biomart, as described in the Materials and methods. This analysis was used to generate the data presented in *Figure 2—figure supplement 2A–C* and *Supplementary files 4* and *5*.

• Supplementary file 4. Table containing the order of heatmap genes and values for *Figure 2—figure supplement 2B*. Genes displayed in the heat map depicting the nRNASeq data of this study were selected as follows: genes with a logFC (Cherry$^+$ vs Cherry$^-$)>0, padj (Cherry$^+$ vs Cherry$^-$)<0.05 and an average TPM of 3, and genes enriched in neurons in the *Zeisel et al., 2018* dataset (logFC neuron vs. glia >0.2 and p-value<0.05). This analysis identifies 366 mouse ENS neuron-enriched genes that have orthologues present in our zebrafish Cherry$^+$ transcriptome dataset, including *phox2bb*, *ret*, *elavl3*, *elavl4*, *prph*, *vip*, *nos1*, and likely reflect the neuronal component of our bulk dataset (See also *Figure 2—figure supplement 2A,B*).

• Supplementary file 5. Table containing the order of heatmap genes and values for *Figure 2—figure supplement 2C*. Genes displayed in the heat map depicting the nRNASeq data of this study were selected as follows: genes with a logFC (Cherry$^+$ vs Cherry$^-$)>0, padj (Cherry$^+$ vs Cherry$^-$)<0.05 and an average TPM of 3, and genes enriched in neurons in the *Zeisel et al., 2018* dataset (logFC glia vs. neurons > 0.2 and p-value≤0.05). This analysis identifies 63 mouse ENS glia-enriched genes that have orthologues present in our zebrafish Cherry$^+$ transcriptome dataset, including *sox10*, *foxd3*, *plp1b*, *zeb2b*, *vim* and *sox2*. Significantly we do not observe canonical glial markers *gfap*, *s100* and *fabp7*. Y and Z. (See also *Figure 2—figure supplement 2A,C*).

• Supplementary file 6. Table containing the order of heatmap genes and values for *Figure 2—figure supplement 1H*. Genes displayed in the heat map depicting the nRNASeq data of this study were selected as follows: genes with a logFC (Cherry$^+$ vs Cherry$^-$)>0, padj (Cherry$^+$ vs Cherry$^-$)<0.05 and an average TPM of 3. We removed from this list the genes found in *Supplementary file 1*. This selection highlights 660 genes. Gene names and Ensembl gene IDs found in column K.

• Supplementary file 7. Zebrafish orthologues of the mouse genes identified in Table 1 of *Rao et al., 2015* PMID:26119414 "'Top 25 genes enriched in PLP1+ enteric glia', generated using the ZFIN and Ensembl databases. Column A shows the zebrafish gene names of the orthologues of the mouse genes shown in Column B

• Transparent reporting form

## Data availability

High-throughput sequencing data have been deposited in GEO under accession codes GSE145885.

The following dataset was generated:

| Author(s) | Year | Dataset title | Dataset URL | Database and Identifier |
|---|---|---|---|---|
| Heanue T, Boeing S, Pachnis V | 2020 | Expression analysis of adult zebrafish enteric nervous system | https://www.ncbi.nlm.nih.gov/geo/query/acc.cgi?&acc=GSE145885 | NCBI Gene Expression Omnibus, GSE145885 |

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
