## [Decision Letter]

**Acceptance summary:**

This work investigates the cellular and molecular characteristics of the non-neuronal compartment of the enteric nervous system and identifies both expected and surprising properties of enteric glia cells in teleost fish. Importantly, this study finds that enteric glia cells in teleosts – in contrast to their mammalian counterpart – are neural progenitor cells during development into adulthood. These findings are of great importance for understanding the basic biology of how the enteric nervous system develops and is maintained in the adult, its regenerative potential, and the potential for therapeutic applications using enteric progenitor cells to treat human gut diseases.

**Decision letter after peer review:**

Thank you for submitting your article "Enteric glia as a source of neural progenitors in adult zebrafish" for consideration by *eLife*. Your article has been reviewed by three peer reviewers, one of whom is a member of our Board of Reviewing Editors, and the evaluation has been overseen by Didier Stainier as the Senior Editor. The reviewers have opted to remain anonymous.

The reviewers have discussed the reviews with one another and the Reviewing Editor has drafted this decision to help you prepare a revised submission.

As the editors have judged that your manuscript is of interest, but as described below that additional experiments are required before it is published, we would like to draw your attention to changes in our revision policy that we have made in response to COVID-19 (https://elifesciences.org/articles/57162). First, because many researchers have temporarily lost access to the labs, we will give authors as much time as they need to submit revised manuscripts. We are also offering, if you choose, to post the manuscript to bioRxiv (if it is not already there) along with this decision letter and a formal designation that the manuscript is 'in revision at *eLife*'. Please let us know if you would like to pursue this option. (If your work is more suitable for medRxiv, you will need to post the preprint yourself, as the mechanisms for us to do so are still in development.)

Summary:

Enteric neurogenesis is an area of intense interest and ongoing debate. At least some of this debate has been caused by discrepancies among data generated from animal models designed to identify neural precursor cells in mice. This question is of great importance both for understanding the basic biology of how the enteric nervous system develops and is maintained in the adult and for potential therapeutic applications, which aim to use progenitor cells to treat ENS diseases. The study is done in the zebrafish model system, in which the non-neuronal compartment of the ENS has not been characterized. Zebrafish are an attractive model system to tackle this question as they have widespread adult neurogenesis in other parts of the nervous system. The authors utilize molecular marker analysis, transcriptional profiling, and live imaging to probe the biology of glia in the zebrafish ENS. This work is a substantial and important contribution because there is little known about enteric glia in zebrafish, and the mechanisms of adult ENS neurogenesis are also poorly understood. The manuscript is well-written and well-illustrated. However, all three reviewers felt that there are major concerns with the conclusions and interpretation of the data with the main concern being that the evidence suggesting that the neural progenitor cells in zebrafish are enteric glia is insufficient and that this conclusion is premature based on the data.

Essential revisions:

1) The characterization of the non-neuronal compartment is done in two ways: first using an enhancer trap Gal4 driver line crossed to *UAS:GFP*. At 7 dpf, GFP^+^ cells are colocalized with Elavl. The interpretation that the GFP labels ENS progenitors and their descendants is not completely convincing. GFP will label the offspring for only a short time, except if the *SAGFF234A* transgene continues to be expressed in non-neuronal offspring, which has not been shown. Therefore, this experiment does not convey a permanent lineage label – the non-neuronal compartment could be much larger if the non-neuronal cells that are not progenitor cells do not express *UAS:GFP*.

Secondly, the lineage tracing using the *Cre*/*loxP* system performed relies on the fact that the sox10:Cre line and the reporter line labels the entire lineage of the ENS. Even though the sox10:Cre line has been published as labeling all neural-crest derivatives, the reporter line that was used in the original paper uses a different minimal promoter than the reporter line used in this study. Besides, Figure 1A shows quite a few Elavl positive cells that are not positive for mCherry and even in the adult, not all Elavl cells are mCherry positive. This shows that this approach does not even label all neuronal cells, which raises the possibility that also the non-neuronal compartment is not completely labeled.

To show that the mCherry^+^ cells fully encompass the entire ENS lineage, the authors have to explain why there are Elavl^+^/mCherry^-^ cells. If this cannot be explained, a different Cre driver line needs to be used to label the entire ENS lineage.

2) The evidence provided is not sufficient to conclude that the neural progenitor cells in zebrafish are enteric glia. The data do provide strong evidence that these cells are neural stem cells, but the evidence that they are glia is weak. There is no functional assessment of the cells and the conclusion that they are glia is based solely on morphology and limited overlap in gene expression with mammalian glia. The non-neuronal compartment of the zebrafish ENS does seem to share some properties with enteric glia and stem cells, but the same is true for many cells including fibroblasts and neurons. It may be more accurate to say that mammalian enteric glia share some genomic overlap with neural stem cells in zebrafish than vice versa. The cells identified in this report do not express typical glial markers and are very different in terms of their morphology (see points b and c below). So far, the reviewer's opinion is that the simplest conclusion based on the data provided is that the cells identified here are neural stem cells, but there is little definitive evidence that they are glia. Concluding that the cells are glia in the Results section and referring to the cells as glia throughout the manuscript seems premature and inappropriate. These interpretations could be discussed in relation to the other potential possibilities in the Discussion section. The reviewers have brought up several points listed below that need to be addressed to provide more convincing data to show that the cells identified in the study are enteric glia.

a) There is no functional assessment of the cells and the conclusion that they are glia is based solely on morphology and limited overlap in gene expression with mammalian glia. The non-neuronal compartment of the zebrafish ENS does seem to share some properties with enteric glia and stem cells, but the same is true for many cells including fibroblasts and neurons. It may be more accurate to say that mammalian enteric glia share some genomic overlap with neural stem cells in zebrafish than vice versa. The cells identified in this report do not express typical glial markers and are very different in terms of their morphology (see points b).

b) The morphology of the cells identified in this study does not seem entirely consistent with that of enteric glia in mammals. As shown in Figure 3, GFP^+^ cells are very large (even larger than the neurons) and have extensive long branches. Data in Figure 7—figure supplement 1 show that GFP^+^ cells have processes that extend several hundred microns and this is not consistent with the morphology of mammalian enteric glia. The authors suggest that the GFP^+^ cells identified here correspond to the four morphological subtypes of enteric glia identified in mammals (see work by Hannani for original identification). How this can be true is not clear since two of the four subtypes of mammalian enteric glia are identified based on their localization within ganglia or interganglionic nerve bundles and the zebrafish ENS lacks ganglia. These points need to be addressed in the manuscript.

c) One of the major strengths of the nRNAseq experiments is that this strategy avoids some of the potential issues of cellular damage associated with tissue dissociation and cell isolation for RNA sequencing. Based on this, it is not clear why the authors chose to compare their transcriptional data with the mouse dataset generated by Rao et al. that used the same lengthy and damaging cellular isolation protocols that the authors were trying to avoid. More comparable datasets for mice are available such as the dataset generated by Delvalle et al., 2018, that used the RiboTag method to isolate glial specific translating RNA for sequencing. Similarly, single-cell transcriptional data by Zeisel et al., 2018, is available for mouse enteric glia. Another concern with using the Rao et al. dataset is that it contains a significant amount of contamination from neurons. These issues make the Rao et al. dataset less appropriate for comparisons in this study than other available datasets. In addition, it is not clear why the authors compare their results from RNA-seq of mCherry^+^ cells just with the 25 most highly expressed genes in the data set of Rao et al. in PLP1^+^ enteric glia and not with their entire data set. Also, even though some of the genes of the PLP1 data set are enriched, a portion of them are enteric progenitor markers (e.g. *foxd3*, *sox10*). Also, these genes could be expressed in different cell subpopulations, which has not been addressed by the authors – is the *her4.3:EGFP* cell population maybe an enteric progenitor cell population and an enteric glia cell population? It is important to provide much more extensive confirmation of gene expression in her4.3EGFP^+^ cells, particularly with more "bona fide" glial markers, such as PLP1, instead of only *in situ*s for *foxd3* and *sox10* (which are also markers for enteric progenitor cells) and *sox2* (which in the zebrafish CNS is expressed in progenitor cells and neurons). Also, colocalization between those markers would be important to test if the *her4.3:gfp* cell population consists of one cell type or different cell populations.

d) In lieu of using any of these empirically identified markers, the authors utilize a Notch activity-driven GFP transgenic reporter *Tg(her4.3:EGFP)* to identify putative enteric glia in all further experiments. They go on to conclude that all GFP^+^ cells in this transgenic line are enteric glia. This would be more convincing if: (1) the authors showed what proportion of GFP^+^ cells in the *Tg(her4.3:EGFP)Sox10Cre:Cherry* fish are Cherry^+^ and (2) to what extent Notch pathway targets were enriched in the non-neuronal Cherry^+^ cells from the RNA sequencing experiment.

3) A major claim is that "Notch signaling regulates neuronal differentiation". This is based on qualitative observations of occasional cells that were GFP^+^/Hu^+^ in the larval fish ENS and distinct studies in adult fish (EdU pulse chase in Figure 6C and LY411575 administration to sox10Cre;Cherry animals, Figure 7). Their results show that there is a marked increase in the number of non-neuronal cells incorporating EdU upon Notch pathway inhibition, suggesting glial proliferation, and a much smaller expansion in the number of Hu^+^ cells that are EdU^+^. In the absence of lineage tracing, this is insufficient evidence to conclude that glia differentiate into neurons upon inhibition of Notch. There may be a small pool of *her4.3:EGFP*^+^ progenitors that are not glia but can give rise to neurons, glia or both. Alternatively, a small number of neurons might re-enter the cell cyle upon Notch inhibition. In either case, the effect seems to be on proliferation rather than differentiation unless the authors have additional experimental data not presented. Live imaging of fish transgenic for both a neuronal reporter and *her4.3:EGFP*, that could visualize *her4.3:EGFP*^+^ cells turning off GFP as they upregulate a neuronal marker in a Notch inhibitor-dependent manner would be stronger evidence.

4) The conclusions from the MCM5 labeling and BrdU data in Figures 6, Figure 6—figure supplement 1 and 2 are not clear. If approximately 10% of the GFP^+^ cells are proliferative (positive for MCM5), how are only 10% positive for BrdU? Proliferative cells and their progeny should all be positive for BrdU so one would expect the percentage of BrdU^+^ GFP cells to be much higher. Some of these cells differentiate and lose their GFP signal, but one wouldn't expect this to happen for all cells if these are actually enteric glia. These data would suggest that all proliferative GFP^+^ cells eventually develop into neurons and there is no ongoing gliogenesis since the same number of GFP^+^ cells are labeled by both BrdU and MCM5.

5) Appropriate controls need to be included to validate the specificity of the antibodies used and the reporter lines.

[Editors' note: further revisions were suggested prior to acceptance, as described below.]

Thank you for resubmitting your work entitled "Enteric glia as a source of neural progenitors in adult zebrafish" for further consideration by *eLife*. Your revised article has been evaluated by Didier Stainier (Senior Editor) and a Reviewing Editor.

The reviewers agree that the paper is excellent and has very strong evidence showing that the cells in question are progenitor cells that share characteristics with mammalian enteric glia. The reviewers have identified a remaining issue that needs to be addressed before acceptance, as outlined below:

The reviewers have discussed at length if they think it is appropriate to call the identified cells of the non-neuronal compartment as enteric glia within the Results section. The reviewers think the authors make a good case for calling the cells enteric glia, but feel that the finding that these cells show characteristics of enteric glia cells should be included as the author's interpretation in the Discussion section instead of calling the cells "enteric glia" already in the Results section.

The reviewers suggest either adapting the term "glial-like neural progenitors" or glial-like neural stem cells", as the authors themselves suggested or calling them "putative enteric glia" or "putative glial cells" when referencing the cells in the Results section.

---

## [Author Response]

Essential revisions:1) The characterization of the non-neuronal compartment is done in two ways: first using an enhancer trap Gal4 driver line crossed to UAS:GFP. At 7 dpf, GFP^+^ cells are colocalized with Elavl. The interpretation that the GFP labels ENS progenitors and their descendants is not completely convincing. GFP will label the offspring for only a short time, except if the SAGFF234A transgene continues to be expressed in non-neuronal offspring, which has not been shown. Therefore, this experiment does not convey a permanent lineage label – the non-neuronal compartment could be much larger if the non-neuronal cells that are not progenitor cells do not express UAS:GFP.

We agree with the reviewer’s view that the *SAGFF234A;UAS:GFP* line is not a lineage tracer of ENS progenitors and it was never meant to be used as such. Given the shortage of pan-ENS genetic labelling tools in zebrafish, it only provided an initial estimate of the relative size of the neuronal and non-neuronal components of the intestinal neural networks. To avoid potential confusion we have edited the original text to: “*SAGFF234A;UAS:GFP* animals in which ENS progenitors and enteric neurons were labelled with GFP”. The initial observations with this line were subsequently validated using the sox10Cre;Cherry transgenic combination, which permanently labels the nuclei of early neural crest cells and their descendants.

Secondly, the lineage tracing using the Cre/loxP system performed relies on the fact that the sox10:Cre line and the reporter line labels the entire lineage of the ENS. Even though the sox10:Cre line has been published as labeling all neural-crest derivatives, the reporter line that was used in the original paper uses a different minimal promoter than the reporter line used in this study. Besides, Figure 1A shows quite a few Elavl positive cells that are not positive for mCherry and even in the adult, not all Elavl cells are mCherry positive. This shows that this approach does not even label all neuronal cells, which raises the possibility that also the non-neuronal compartment is not completely labeled.To show that the mCherry^+^ cells fully encompass the entire ENS lineage, the authors have to explain why there are Elavl^+^/mCherry^-^ cells. If this cannot be explained, a different Cre driver line needs to be used to label the entire ENS lineage.

We agree that *Tg(sox10Cre;Cherry)* does not mark *all* ENS cells, and that the labelling efficiency is variable from animal to animal (as is the case for most *Cre/LoxP*-based genetic labelling systems). In the revised manuscript we provide new immunostaining images for Figure 1A and B, which reflect more accurately the proportion of HuC/D^+^Cherry^-^ cells observed in the corresponding quantifications of Figure 1C.

Please note that even the Cre/reporter combination (*Tg(sox10:Cre;ef1a:loxP-GFP-loxP-DsRed2)*) used in the original paper does not mark *all* ENS cells. We specifically describe Cre/reporter efficiencies in the revised text. We provide a new data figure which shows that the labelling of HuC/D^+^ neurons in the gut of 7 dpf larvae from *Tg(sox10Cre;Cherry)* transgenics is 47.1% ± 19.9, whereas the corresponding figure from *Tg(sox10:Cre;ef1a:loxP-GFP-loxP-DsRed2)* animals is 80.8% ± 7.8 (Figure 1—figure supplement 1C). Importantly, these reporters show no differential bias towards neuronal vs. non-neuronal lineages. We support this statement by providing a new data figure (Figure 1—figure supplement 1D), which shows that the proportion of HuC/D^+^ and HuC/D^-^ cells in the ENS of *Tg(sox10Cre;Cherry)* (84.8% ± 7.7 and 15.2% ± 7.7, respectively) and *Tg(sox10:Cre;ef1a:loxP-GFP-loxP-DsRed2)* (86.8% ± 6.4 and 13.2% ± 6.4%) lines are equivalent (p = 0.78). This new data provides independent support to the view that the non-neuronal compartment of zebrafish ENS is considerably smaller relative to its mammalian counterpart.

Our choice of the *Tg(sox10Cre;Cherry)* line was dictated by the fact that it provides clear nuclear labelling and allows combination with other transgenic reporters, such as *Tg(her4.3:EGFP)*. In contrast, the *Tg(sox10:Cre;ef1a:loxP-GFP-loxP-DsRed2)* line renders the entire embryo GFP^+^, making combination with other GFP transgenics impossible. Taken together, we believe that the experiments described in our manuscript, the clarifications provided in this response and new supporting figures strongly support our conclusion that the nonneuronal compartment of the zebrafish ENS is considerably smaller relative to its counterpart in mammals. These findings parallel the glia to neuron ratio in the zebrafish pallium and in the Discussion we provide an evolutionary perspective of this observation.

2) The evidence provided is not sufficient to conclude that the neural progenitor cells in zebrafish are enteric glia. The data do provide strong evidence that these cells are neural stem cells, but the evidence that they are glia is weak. There is no functional assessment of the cells and the conclusion that they are glia is based solely on morphology and limited overlap in gene expression with mammalian glia. The non-neuronal compartment of the zebrafish ENS does seem to share some properties with enteric glia and stem cells, but the same is true for many cells including fibroblasts and neurons. It may be more accurate to say that mammalian enteric glia share some genomic overlap with neural stem cells in zebrafish than vice versa. The cells identified in this report do not express typical glial markers and are very different in terms of their morphology (see points b and c below). So far, the reviewer's opinion is that the simplest conclusion based on the data provided is that the cells identified here are neural stem cells, but there is little definitive evidence that they are glia. Concluding that the cells are glia in the Results section and referring to the cells as glia throughout the manuscript seems premature and inappropriate. These interpretations could be discussed in relation to the other potential possibilities in the Discussion section. The reviewers have brought up several points listed below that need to be addressed to provide more convincing data to show that the cells identified in the study are enteric glia.a) There is no functional assessment of the cells and the conclusion that they are glia is based solely on morphology and limited overlap in gene expression with mammalian glia. The non-neuronal compartment of the zebrafish ENS does seem to share some properties with enteric glia and stem cells, but the same is true for many cells including fibroblasts and neurons. It may be more accurate to say that mammalian enteric glia share some genomic overlap with neural stem cells in zebrafish than vice versa. The cells identified in this report do not express typical glial markers and are very different in terms of their morphology (see points b).b) The morphology of the cells identified in this study does not seem entirely consistent with that of enteric glia in mammals. As shown in Figure 3, GFP^+^ cells are very large (even larger than the neurons) and have extensive long branches. Data in Figure 7—figure supplement 1 show that GFP^+^ cells have processes that extend several hundred microns and this is not consistent with the morphology of mammalian enteric glia. The authors suggest that the GFP^+^ cells identified here correspond to the four morphological subtypes of enteric glia identified in mammals (see work by Hannani for original identification). How this can be true is not clear since two of the four subtypes of mammalian enteric glia are identified based on their localization within ganglia or interganglionic nerve bundles and the zebrafish ENS lacks ganglia. These points need to be addressed in the manuscript.

We are pleased that the reviewers are satisfied with our conclusion that *her4.3:EGFP*^+^ cells have properties of neural stem cells. However, they argue that we do not provide enough evidence to support the idea that these cells represent the elusive enteric glia of zebrafish. We disagree with their view and the statement in the review summary that the *her4.3:EGFP*^+^ cells “…are very different in terms of their morphology.” relative to canonical mammalian enteric glia. On the contrary, considering the evolutionary distance between teleosts and mammals, the distinct organization of zebrafish and mouse ENS and the documented plasticity of glial cells (including enteric glia), we would argue that in terms of morphology, the *her4.3:EGFP*^+^ cells *are remarkably similar to mammalian enteric glia*. Given the lack of a diagnostic morphological criterion(a) for enteric glia, such an argument is based inescapably on a list of criteria described in our manuscript. Specifically:

Our CLEM analysis (Figure 4 and Figure 4—figure supplement 1) demonstrates that the *her4.3:EGFP*^+^ cells share the majority of the ultrastructural characteristics of enteric glia, as they have been described in the seminal publications of G. Gabella. Among them are: (1) the close association and specialised contact points between *her4.3:EGFP*^+^ cells with enteric neurons, (2) the partial enveloping of enteric neurons by *her4.3:EGFP*^+^ processes, (3) the partial enveloping of axonal bundles and their subdivision into sectors by *her4.3:EGFP*^+^ cellular processes, (4) the scarcity of cytoplasm, and (5) the presence of nuclear crenations, which are observed almost universally in all glial cells in invertebrates and vertebrates.

Regarding the size of *her4.3:EGFP*^+^ cells, Figure 3C shows clearly that the nuclear size of *her4.3:EGFP*^+^ cells is comparable (and often smaller) to that of nearby neuronal nuclei. For example, compare the GFP^+^Cherry^+^ nuclei (arrows) to the GFP^-^Cherry^+^ nuclei (arrowheads). In the revised manuscript we make a note of this feature in the Figure 3C legend. The similar (or smaller) nuclear size of *her4.3:EGFP*^+^ cells in combination with the sparsity of their cytoplasm (revealed by the CLEM analysis, Figure 4) clearly indicates that their cell bodies are not larger in comparison to neurons. In our view, light microscopy of cellular networks (such as glia networks) is not the best approach to assess the overall size of individual cells, due to intermingling of cellular processes (see for example the network of GFP^+^ cells in Figure 3A and in higher magnification in Figure 3C, arrows). For this reason, both figures mentioned in your summary (Figure 3 and Figure 7—figure supplement 1), do not address the question of size of *her4.3:EGFP*^+^ cells. We have included a single z image as an inset into Figure 7—figure supplement 1B that clarifies that the GFP immunostainings interpreted by the reviewers as revealing cell processes extending for several hundred microns are actually groups of adjacent cells, and amended the figure legend accordingly.

However, our CLEM analysis has enabled us to compare the size of *her4.3:EGFP*^+^ cells and enteric neurons and measure the length of *her4.3:EGFP*^+^ cell processes (and thus estimate their size/volume) more accurately. Figure 4 and Video 1, which show *her4.3:EGFP*^+^ cells and Cherry^+^ neurons in close association, indicate that neurons are larger than the nearby GFP^+^ cells. We support this observation with specific measurements through the CLEM dataset. Specifically, the dimensions of the somata of the GFP^+^ cells shown in Figure 4A/B, Figure 4C/D and Figure 4—figure supplement 1B,C are: 8x7.6x2.5 µm (=79.6 µm^3^), 10x6x3 µm (=94.3 µm^3^) and 7.75x6.5x3 µm (=79.1 µm^3^), respectively (see also Materials and methods). Individual projections were also measured and are either shorter sheets that extend to approximately 4 µm from the middle of the cell body or they are longer extensions up to 18 µm. Specifically, the GFP^+^ cell in 4A/B has two processes of 4 µm and a longer process of 18 µm while the GFP^+^ cell in Figure 4—figure supplement 1B,C has two longer extensions of 8.5 and 9 µm. Cell sizes of mmCherry^+^ neurons have also been measured for reference. The Cherry^+^ cell somata in Figure 4A (left) and Figure 4A (right) are approximately 8.5x11.2x8 µm (=398.8 µm^3^) and 10x4.5x9.75 µm (=229.7 µm^3^), respectively, with projection lengths ranging from 16 to 55 µm. Taken together, these measurements support the view that, *similar to mammalian enteric glia*, *her4.3:EGFP*^+^ cells in zebrafish gut are highly branched, with shorter sheet like processes that extend 5 µm and longer projections that extend up to 18 µm. We confirm that these cells do *not* extend several hundred microns, as suggested, and by comparison are significantly smaller to neurons. Our revised manuscript includes these measurements in the text and the corresponding figure legends, with measurement methods described in the Materials and methods section.

These measurements allow us also to compare the size of *her4.3:EGFP*^+^ cells to that of mammalian enteric glia. The study of (Hanani and Reichenbach, 1994) indicates that the volume of guinea-pig enteric glia is approximately 330 µm^3^ and the average process length is ~29 µm (Type I “protoplasmic” glia) and ~83 µm (Type II “fibrous” glia), with the longest process 115 µm long. Boesmans et al. describe 4 types of mouse glia (Types I-IV), with cell bodies ranging from ~241 µm^3^ – ~335 µm^3^, and average process length of 30 µm to 124 µm, with the longest process length found in Type IV glia (138 µm). This allows us to conclude that *her4.3:EGFP*^+^ cells are smaller in comparison to mammalian enteric glia, which is consistent with the fact that zebrafish cells are typically smaller relative to mammalian cells.

The reviewers also argue that because of the lack of typical enteric ganglia in zebrafish, one would not have expected to detect Type I and Type II enteric glia, which are “defined by their localization within ganglia or interganglionic nerve bundles”. We disagree with this view. First, our qualitative light microscopy analysis demonstrates that *her4.3:EGFP*^+^ cells with Type I and Type II morphologies are observed (Figure 4—figure supplement 1B and C, respectively). Secondly, mammalian Type I enteric glia are associated with neuronal cell bodies (within ganglia), and we demonstrate in Figure 3—figure supplement 1B that a proportion of *her4.3:EGFP*^+^ cells are similarly associated with enteric neuronal cell bodies, with multiple processes wrapping around neuronal somata. On the other hand, mammalian Type II enteric glia are associated with neuronal projections (in interganglionic regions), and we demonstrate in Figure 3—figure supplement 1C that a proportion of *her4.3:EGFP*^+^ cells are similarly associated with enteric neuron processes. We argue that these clear morphological and contact specialisations are not surprising, but rather *expected* from zebrafish enteric glia, which, in a manner *analogous* to their mammalian counterparts, must establish close anatomical and functional interactions with individual (or small clusters of) enteric neurons (Type I) and their projections (Type II). In the revised manuscript, we have extended our description of her4.3EGFP^+^ cell morphologies, to further support the parallels we draw to the 4 types of mouse enteric glia and to address the reviewer’s comments. We have also provided new images for Figure 3—figure supplement 1, which we believe make our point of view clearer.

The blending of glia and neural stem cell characteristics within the same cell type is well established in neuroscience. An extensive literature is available on the glia character of neural stem cells and the neural stem cell character of glia cells, in vertebrates and invertebrates, during development and in adult stages. Several investigators have analysed the stem cell properties of Schwann cell precursors associated with growing nerves (Dyachuk et al., 2014; Kaukua et al., 2014), the acquisition of embryonic stem cell properties by Schwann cells (Clements et al., 2017), (Masaki et al., 2013), the neurogenic properties of carotid body glia cells (Pardal et al., 2007), the neural stem cell character of mammalian enteric glia revealed *in vivo* (Belkind-Gerson et al., 2017), (Laranjeira et al., 2011), *in vitro* reprogramming of adult enteric glia from the mouse intestine ((Joseph et al., 2011), (Laranjeira et al., 2011) and our own unpublished observations), and the neural stem cell properties of glia in the fly brain (https://www.biorxiv.org/content/10.1101/721498v1). Highly relevant to our studies is the mixed character of radial glial cells in the zebrafish pallium, which serve as the prototypical neural stem cells of the CNS and at the same time are “standing in” for the missing astrocytes in order to support pallial neurons and circuits. Therefore, given the similarities of *her4.3:EGFP*^+^ cells with mammalian enteric glia (in terms of morphology and gene expression profile) *and the absence of a cell population that expresses canonical glial markers*, we argue that these cells, in addition to their role as enteric neural progenitors, provide also the necessary glia function to intestinal neural circuits. Our Discussion section describes extensively the evidence from the literature that supports the co-existence of glia and neural stem cell characteristics.

We agree with the reviewers that it would have been wonderful to use a diagnostic functional assay for enteric glia. However, it is unclear whether any of the available assays for mammalian enteric glia (such as genetically encoded fluorescent indicators to trace changes in intracellular Ca^2+^ concentration) would be directly transferable to zebrafish and, more importantly, whether they would be diagnostic of enteric glia cell identity in this species. In any case, such an assay is currently unavailable for *in vivo* analysis of enteric glia in *adult* zebrafish and it is of interest to determine whether certain confounding issues can be overcome (such the presence of the liver impeding visibility, even in transparent adult zebrafish). However, we believe that such an undertaking is beyond the scope of the current manuscript.

The topic of constitutive neurogenesis in the vertebrate ENS is hotly debated. We believe that our work makes important contributions to this debate. First, it provides evidence that enteric neurogenesis does take place *in vivo* and second, that one does not have to look to a distinct population of cells that supports neurogenesis in the adult ENS. According to our data, it can be provided by the all-familiar enteric glia.

We have had longstanding debates amongst ourselves as to whether we should be calling the *her4.3:EGFP*^+^ cells of the zebrafish ENS “enteric glia” or not. Ultimately, the accumulated evidence described here of the shared lineage of *her4.3:EGFP*^+^ cells with ENS neurons, their cellular locations and close association with enteric neurons, their distinctive morphologies, their ultrastructural features, their expression of multiple well established enteric glial markers and their neural progenitor properties outweighed the lack of expression of GFAP/*gfap* (not requisite for mammalian glia in any case (Boesmans et al., 2015)), leading us to find “enteric glia” as the best descriptor of this population. We note that the original description of enteric glia from (Gabella, 1981) and (Hanani and Reichenbach, 1994) were based on the criteria of position, morphology and ultrastructural features *alone*, and that the identity of these cells has stood the test of time. Only subsequently have these cells been found to have progenitor cell properties, transcriptionally studied or analysed with electrophysiological and imaging tools to explore functional properties, such as activity. Moreover, we note that a corollary to not calling *these* cells “enteric glia” is to declare that the zebrafish ENS does not contain enteric glia, an idea which is undoubtedly more heretical than a glial cell that does not express GFAP. We also considered whether to propose a different terminology that would also capture their undisputed function as ENS stem cells. However, we do not know whether all *her4.3:EGFP*^+^ cells function as progenitors or whether it is a subpopulation of *her4.3:EGFP*^+^ cells with this property, and thus the general naming of the entire population as “enteric glia” is more accurate and future-proof. So, the fact that we could have chosen to term *her4.3:EGFP*^+^ cells “glial-like neural progenitors” or “glial-like neural stem cells” (GNPs? GNSC?) is clear to us, and ultimately would not detract from our findings, but is ambiguous. We argue, for all the reasons laid out in this rebuttal (in points 2a,b,c, above and below), to retain the name “enteric glia”.

c) One of the major strengths of the nRNAseq experiments is that this strategy avoids some of the potential issues of cellular damage associated with tissue dissociation and cell isolation for RNA sequencing. Based on this, it is not clear why the authors chose to compare their transcriptional data with the mouse dataset generated by Rao et al. that used the same lengthy and damaging cellular isolation protocols that the authors were trying to avoid. More comparable datasets for mice are available such as the dataset generated by Delvalle et al., 2018, that used the RiboTag method to isolate glial specific translating RNA for sequencing. Similarly, single-cell transcriptional data by Zeisel et al., 2018, is available for mouse enteric glia. Another concern with using the Rao et al. dataset is that it contains a significant amount of contamination from neurons. These issues make the Rao et al. dataset less appropriate for comparisons in this study than other available datasets. In addition, It is not clear why the authors compare their results from RNA-seq of mCherry^+^ cells just with the 25 most highly expressed genes in the data set of Rao et al. in PLP1^+^ enteric glia and not with their entire data set. Also, even though some of the genes of the PLP1 data set are enriched, a portion of them are enteric progenitor markers (e.g. foxd3, sox10). Also, these genes could be expressed in different cell subpopulations, which has not been addressed by the authors – is the her4.3:EGFP cell population maybe an enteric progenitor cell population and an enteric glia cell population? It is important to provide much more extensive confirmation of gene expression in her4.3EGFP^+^ cells, particularly with more "bona fide" glia markers, such as PLP1, instead of only *in situs* for foxd3 and sox10 (which are also markers for enteric progenitor cells) and Sox2 (which in the zebrafish CNS is expressed in progenitor cells and neurons). Also, colocalization between those markers would be important to test if the her4.3:GFP cell population consists of one cell type or different cell populations.

We thank the reviewers for their interesting comments and useful suggestions. We do recognise the limitation of the Rao study for statistically robust analysis, including the lack of replicates (1 replicate per condition), which is not mentioned by the reviewers. It was for this reason that we found more in depth use of their data was fraught and we instead relied on the top 25 gene list that was presented by the authors, reasoning that it was the part of their data they felt the most confidence in. Regarding the very interesting work of Delvalle and colleagues, please note that this dataset does not identify glia-specific genes, but rather genes whose expression changes in enteric glia in response to specific experimental manipulation (colitis). Although this analysis is very well placed to identify the “inflammatory” character of mammalian enteric glia, it does not help us explore the similarities between zebrafish and mammalian enteric glia.

We found more fortune however, with the (Zeisel et al., 2018) publication mentioned by the reviewers, despite requiring a comparison between bulk (our) and single cell (Zeisel et al.) data. This analysis has identified an extended list of genes in the zebrafish ENS transcriptome with orthologues expressed in mammalian glia. The new data is described in the Results section of the revised manuscript and presented in a new Figure 2—figure supplement 2 and new Supplementary files 3, 4 and 5. The methodology of the new analysis is described in the Materials and methods section.

Specifically, we first identified those genes from the Zeisel dataset (raw data downloaded from mousebrain.org and analysed as described in the Materials and methods section) that are differentially expressed between mouse ENS neurons and glia and then identified their orthologues in our zebrafish ENS transcriptome (Supplementary file 3). We show that 366 mouse ENS neuron enriched genes have orthologues in our zebrafish transcriptome dataset, including *ret*, *phox2bb*, *elavl3*, *elavl4*, and likely reflect the neuronal component of our bulk dataset (Figure 2—figure supplement 2A,B). We also show that 63 mouse ENS glia enriched genes have orthologues present in our zebrafish transcriptomic data, suggesting that they are expressed in the non-neuronal component of the zebrafish ENS (Figure 2—figure supplement 2A,C), including *sox10*, *foxd3*, *plp1b*, *zeb2b*, and *Sox2*. Significantly, this analysis did not detect canonical glia markers (*gfap*, *s100b* and *fabp7a/b*) in the zebrafish ENS transcriptome.

The reviewers suggest that we use more “bona fide” glia markers, such as PLP1, to characterise the *her4.3:EGFP*^+^ cells, rather than the “progenitor markers” *sox10* and *foxd3*. However, out of the “traditional” markers used widely in the literature for the identification of enteric glia (Gfap, S100b, Plp1, BFABP, Sox10, Sox2, Foxd3), *none of them* defines enteric glia identity. Some may be upregulated in a subset of enteric glia of adult mammals, but *all* also mark neural progenitors and neural stem cells. This is supported by the literature and our own single cell RNA sequencing of mammalian ENS cells at various developmental stages (unpublished). Since PLP1 was specifically mentioned in the review summary, we note that this gene has been used by the Adameyko lab to drive expression of Cre recombinase in early neural crest cells, and therefore this gene is not materially different from the other genes we have examined. Further future analysis of emerging transcriptomic datasets may perhaps identify novel marker genes that uniquely define enteric (and other peripheral) glia, but such marker(s) do not exist at present.

With regards to whether the *her4.3:EGFP*^+^ cell population consists of one cell type or different populations, we now include new data in Figure 3F of our revised manuscript to show that all *her4.3:EGFP*^+^ cells express *Sox2*. In this figure we also show that *foxd3* is expressed in only a proportion of the her4.3:EGFP^+^/Sox2^+^ cells, giving first indication of molecular differences in *her4.3:EGFP*^+^ cells. This new data is described in our revised manuscript. Heterogeneity is also indicated by the fact that only a proportion of *her4.3:EGFP*^+^ cells are proliferative in our experiments. We now make a specific mention to potential heterogeneity of EGCs in the Discussion of our revised manuscript. Further robust exploration of the question of cell type heterogeneity would be best approached using a comprehensive single-cell transcriptomic strategy, an endeavour that we feel is outside of the scope of this current manuscript.

d) In lieu of using any of these empirically identified markers, the authors utilize a Notch activity-driven GFP transgenic reporter Tg(her4.3:EGFP) to identify putative enteric glia in all further experiments. They go on to conclude that all GFP^+^ cells in this transgenic line are enteric glia. This would be more convincing if: (1) the authors showed what proportion of GFP^+^ cells in the Tg(her4.3:EGFP)Sox10Cre:Cherry fish are Cherry^+^ and (2) to what extent Notch pathway targets were enriched in the non-neuronal Cherry^+^ cells from the RNA sequencing experiment.

We used the *Tg(her4.3:EGFP)* line not “in lieu of…empirically identified markers”, but *because* our search for molecular markers and transgenic reporters that are traditionally used in the field of mammalian enteric glia was unsuccessful, a finding corroborated by our initial and subsequent transcriptomic analysis. This, and evidence from the literature, led us to the hypothesis that a Notch activity transgenic reporter could identify enteric glia in zebrafish, an idea that was subsequently validated experimentally. Of course, having in hand a transgenic line that marks enteric glia facilitates the study of this cell population, as it can be combined with other immunohistochemical and transgenic tools and allows the application of techniques such as CLEM. We have modified and expanded the text to clarify this logic.

We clarify in response to point 1 (above) that the lineage tools we have used do not label *all* ENS cells (new Figure 1—figure supplement 1C) and do not show differential bias towards neural vs. non-neural lineages (new Figure 1—figure supplement 1D). As such, no particular significance can be drawn to the presence of *her4.3:EGFP*^+^ cells that are Cherry^+^ or Cherry^-^, a point made clear in the text. In particular, with regards to the *Cherryher4.3:EGFP*^+^ cells, we are confident that they are part of the non-neuronal lineage of the ENS, as *allher4.3:EGFP*^+^ cells express *Sox2*, a gene which in the tunica muscularis of mammalian gut marks exclusively glial cells (this additional data is now presented in Figure 3F). Furthermore, we often find in the gut of *Tg(sox10Cre;Cherry)* fish large domains in which all *her4.3:EGFP*^+^ cells (and HuC/D^+^ cells) are Cherry^+^ (see for example Figure 3C, D), suggesting stochastic expression of the *Tg(sox10Cre;Cherry)* reporter.

We are confident that the *Tg(her4.3:EGFP)* transgene reflects Notch activity, since inhibition of the Notch signalling using LY411575 silences expression of the transgene (Figure 7—figure supplement 1). In our revised manuscript we have included a transcriptomic results table as a new Supplementary file 1, which highlights differential expression in the Cherry^+^ and Cherry^+^ population, without using any database comparisons. However, the Notch receptors (*notch1a*, *notch1b*, *Notch2*, *notch3*) are not significantly upregulated in either the Cherry^+^ or Cherry^-^ populations, with similar results for multiple Notch targets (all *her* and *hey* genes with the exception of *her2, her4.5* and *her4.4*). This result is not entirely surprising. Given widespread expression of Notch signalling components in multiple tissues of the mammalian adult gut (i.e. lamina propria, blood vessels, epithelia and enteric nervous system; Sander and Powell, 2004, Okamura and Saga, 2008), we expect *notch* receptors and their downstream targets to be expressed in both Cherry^+^ and Cherry^-^ populations. In such case, our bulk differential transcriptomic strategy is not best placed to identify the Notch targets acting within adult glia, and we will not use this experiment to make statements about expression of Notch targets. This question would be best addressed using a single-cell transcriptomic strategy, an undertaking we feel is outside the scope of this current manuscript.

3) A major claim is that "Notch signaling regulates neuronal differentiation". This is based on qualitative observations of occasional cells that were GFP^+^/Hu^+^ in the larval fish ENS and distinct studies in adult fish (EdU pulse chase in Figure 6C and LY411575 administration to sox10Cre;Cherry animals, Figure 7). Their results show that there is a marked increase in the number of non-neuronal cells incorporating EdU upon Notch pathway inhibition, suggesting glial proliferation, and a much smaller expansion in the number of Hu^+^ cells that are EdU^+^. In the absence of lineage tracing, this is insufficient evidence to conclude that glia differentiate into neurons upon inhibition of Notch. There may be a small pool of her4.3:EGFP^+^ progenitors that are not glia but can give rise to neurons, glia or both. Alternatively, a small number of neurons might re-enter the cell cyle upon Notch inhibition. In either case, the effect seems to be on proliferation rather than differentiation unless the authors have additional experimental data not presented. Live imaging of fish transgenic for both a neuronal reporter and her4.3:EGFP, that could visualize her4.3:EGFP^+^ cells turning off GFP as they upregulate a neuronal marker in a Notch inhibitor-dependent manner would be stronger evidence.

Thank you for the opportunity to clarify the issues related to this comment.

First, our evidence that Notch signalling regulates neuronal differentiation is based exclusively on the data presented in Figure 7, which shows that upon exposure to the LY411575 inhibitor, both the percentage of nonneuronal ENS cells that incorporate EdU (Cherry^+^Hu^-^ EdU^+^) and the percentage of EdU-labelled enteric neurons (Cherry^+^Hu^+^EdU^+^) increases at two adult stages (3 mo and 6 mo). Please note that in these experiments we do not follow specifically *her4.3:EGFP*^+^ cells because Notch inhibition silences the transgene (Figure 7—figure supplement 1) and therefore we are unable to use it to follow *her4.3:EGFP*^+^ cells. We agree with the reviewers that we do not provide evidence that Notch regulates directly neuronal differentiation in the adult zebrafish ENS and that the increase in the fraction of EdU-lableled Hu^+^Cherry^+^ enteric neurons following treatment with LY411575 could result from the increase in the size of the neurogenic progenitor pool. We have now revised our manuscript to reflect more accurately our experimental results and indicate that Notch signalling regulates the dynamics of non-neuronal cells.

Second, our conclusion that *her4.3:EGFP*^+^ cells give rise to neurons during homeostasis has no bearing on the identified role of Notch signalling in neuronal differentiation. We simply aim to show that there is constitutive neurogenesis in the ENS of adult zebrafish and that the GFP^+^ non-neuronal cell population is a source of newborn enteric neurons. Suggestive evidence for neuronal differentiation of *her4.3:EGFP*^+^ cells was first provided by the occasional double labelling of individual cells by Hu and GFP (Figure 5—figure supplement 1D) and the live imaging presented in Figure 5. Further direct evidence was provided by the pulse-chase experiments (shown in Figure 6A-E) *in conjunction with the computational model* presented in Figure 6F, G. Thymidine analogue incorporation has been used widely in neuroscience to demonstrate de novo neurogenesis in different parts of the adult nervous system. Taking advantage of the postmitotic nature of enteric neurons, we used EdU pulse-labelling to identify EdU^+^ neurons in the gut, which provided unequivocal evidence for constitutive neurogenesis in the adult zebrafish ENS. Furthermore, *spatial analysis* of EdU^+^*her4.3:EGFP*^+^ and EdU^+^Hu^+^ cells established that *her4.3:EGFP*^+^ cells serve as precursors of enteric neurons. We recognise that the use of a *Cre/LoxP*-based lineage tracing system (which we have used extensively in our laboratory) would have been an alternative way to demonstrate neuronal differentiation of *her4.3:EGFP*^+^ cells, but so far we have been unable to identify a suitable transgenic line that would allow us to conduct this experiment in adult zebrafish. While such a line would be useful in the future to explore the kinetics of neuronal differentiation of *her4.3:EGFP*^+^ cells in the adult ENS, we believe that the current studies provide strong evidence for constitutive neurogenesis and identifies these cells as a source of newborn enteric neurons throughout life. Indeed, in the review summary the reviewers are supportive of the neural stem cell properties our studies demonstrate.

4) The conclusions from the MCM5 labeling and BrdU data in Figures 6, Figure 6—figure supplement 1 and 2 are not clear. If approximately 10% of the GFP^+^ cells are proliferative (positive for MCM5), how are only 10% positive for BrdU? Proliferative cells and their progeny should all be positive for BrdU so one would expect the percentage of BrdU^+^ GFP cells to be much higher. Some of these cells differentiate and lose their GFP signal, but one wouldn't expect this to happen for all cells if these are actually enteric glia. These data would suggest that all proliferative GFP^+^ cells eventually develop into neurons and there is no ongoing gliogenesis since the same number of GFP^+^ cells are labeled by both BrdU and MCM5.

We thank the reviewers for the opportunity to clarify these points. We believe that the similar size of the MCM5^+^ and EdU^+^ subsets of *her4.3:EGFP*^+^ cells is explained by the cell cycle dynamics of this cell population. *Mcm5* is expressed at varying levels throughout the cell cycle while an antibody that binds to the MCM5 protein (and used widely in the field and in our study (Chapouton et al., 2010; Ryu et al., 2005)) labels the G1/S transition phase (whose length depends on how rapidly cells proliferate) and certain stages of G2. On the other hand, EdU marks cells that go through S phase. In quiescent tissues, one would expect that over a 3-day EdU pulse proliferating cells would have gone through the various phases of the long cell cycles only once, resulting in similar fractions of MCM5^+^ and EdU^+^ cells. Indeed, given that MCM5 labels several stages of the cell cycle, including G1/S (which in quiescent tissues can be exceptionally long), it could even label a greater population of cells relative to EdU. We note that examples of similar numbers of cells labelled with MCM5^+^ and EdU^+^ in proliferating tissues have also been reported in the literature (Lange et al., 2020).

Our initial attempts to EdU label the *her4.3:EGFP*^+^ cells using 1-hour pulses (a protocol used to identify highly proliferative tissues) failed to label enough cells to study effectively their proliferation/neurogenesis and suggested that we are dealing with a cell population which, similar to RGCs in the pallium, is generally quiescent and any dividing cells have long cell cycles. Based on these observations, we reasoned that longer time frames would allow us to capture the *her4.3:EGFP*^+^ cells that progress through and complete the cell cycle and therefore we adopted the 3-day EdU pulse protocol. Our finding that the majority of her4.3:EGFP^+^EdU^+^ cells at the end of the 3-day EdU pulse (t0) are in doublets supports the view that during this period any cycling cells have undergone only one cell division and explains why we detect similar fractions of MCM5^+^ and EdU^+^ cells. In our revised manuscript we compare our MCM5 and EdU results and state our conclusion that *Tg(her4.3:EGFP)*^+^ cells represent a largely quiescent cell population and the that dividing cells have long cell cycles.

Despite the neuronal differentiation observed over the 11-day chase period, a significant fraction of *her4.3:EGFP*^+^ cells in 3 month old zebrafish retain their EdU label (Figure 6D), suggesting that proliferating glial cells return to quiescence and their glial role (=gliogenesis). Similar observations were made also for 6-month-old animals.

In future studies we plan to characterize in greater detail the proliferation and differentiation dynamics of *her4.3:EGFP*^+^ cells.

5) Appropriate controls need to be included to validate the specificity of the antibodies used and the reporter lines.

The antibodies and reporter lines applied in this study are all widely used, and we provide appropriate references and/or catalogue numbers. For many antibodies and reporter lines, particularly for those that we show negative data in the ENS (i.e S100B/BFABP/GFAP/*Tg(gfap:GFP)*) we show larval spinal cord staining as internal positive controls in Figure 1—figure supplement 1 (J-O). We clarify this point in the Material and methods section to indicate that larval CNS and adult brain are used as positive controls in our experiments, and explicitly refer to this figure. We have also clarified in the Materials and methods section that “no primary antibody” negative control experiments have been performed for immunostaining, and yielded negative results.

[Editors' note: further revisions were suggested prior to acceptance, as described below.]

The reviewers agree that the paper is excellent and has very strong evidence showing that the cells in question are progenitor cells that share characteristics with mammalian enteric glia. The reviewers have identified a remaining issue that needs to be addressed before acceptance, as outlined below:The reviewers have discussed at length if they think it is appropriate to call the identified cells of the non-neuronal compartment as enteric glia within the Results section. The reviewers think the authors make a good case for calling the cells enteric glia, but feel that the finding that these cells show characteristics of enteric glia cells should be included as the author's interpretation in the Discussion section instead of calling the cells "enteric glia" already in the Results section.The reviewers suggest either adapting the term "glial-like neural progenitors" or glial-like neural stem cells", as the authors themselves suggested or calling them "putative enteric glia" or "putative glial cells" when referencing the cells in the Results section.

We are pleased that our revised manuscript now makes “a good case for calling the cells enteric glia”. The only issue remaining is deciding the appropriate place in the manuscript to present the conclusion that the *her4.3:EGFP*^+^ non-neuronal population can be called “enteric glia”.

We have modified our manuscript in accordance with the proposed naming of these cells as “putative enteric glia” in the Results section and “enteric glia” in the Discussion section, after a paragraph outlining our case. On reading this version, we find that calling them “putative enteric glia” in the Results after having presented all the evidence in support of their “enteric glia” character, and then calling them “enteric glia” in the Discussion (without in the meantime presenting any additional evidence), affects the readability of the manuscript, is a source of confusion and blurs the clarity in the overall paper.

We are acutely aware of the need to clearly lay out our case for calling these cells “enteric glia” and appreciate the attention the reviewers have brought to this important point. But at the same time, it’s our wish that the final manuscript’s clarity and readability not be impinged.

We would like to propose an alternative approach to modifying the manuscript, which adopts the reviewers’ suggestion for a summary paragraph, but that we believe would be more effective. We propose that we add a small but clear and evidence-based paragraph at the end of the relevant part of the Results section that would lay out our entire case for calling the *her4.3:EGFP*^+^ cells “enteric glia”. This paragraph would follow the last piece of data that led us to this conclusion, the conventional place for summarizing all relevant evidence. Please note that up to this point we have not used the term “enteric glia” to identify the *her4.3:EGFP*^+^ cells and the term is only used in the second part of the Results section, after all supporting evidence has been presented. We have now modified our manuscript according to this approach as well.